# Molecular structure and interactions within amyloid-like fibrils formed by a low-complexity protein sequence from FUS

Myungwoon Lee[1], Ujjayini Ghosh[1], Kent R. Thurber[1], Masato Kato[2,3] & Robert Tycko [1✉]

Protein domains without the usual distribution of amino acids, called low complexity (LC) domains, can be prone to self-assembly into amyloid-like fibrils. Self-assembly of LC domains that are nearly devoid of hydrophobic residues, such as the 214-residue LC domain of the RNA-binding protein FUS, is particularly intriguing from the biophysical perspective and is biomedically relevant due to its occurrence within neurons in amyotrophic lateral sclerosis, frontotemporal dementia, and other neurodegenerative diseases. We report a high-resolution molecular structural model for fibrils formed by the C-terminal half of the FUS LC domain (FUS-LC-C, residues 111-214), based on a density map with 2.62 Å resolution from cryo-electron microscopy (cryo-EM). In the FUS-LC-C fibril core, residues 112-150 adopt U-shaped conformations and form two subunits with in-register, parallel cross-β structures, arranged with quasi-$2_1$ symmetry. All-atom molecular dynamics simulations indicate that the FUS-LC-C fibril core is stabilized by a plethora of hydrogen bonds involving sidechains of Gln, Asn, Ser, and Tyr residues, both along and transverse to the fibril growth direction, including diverse sidechain-to-backbone, sidechain-to-sidechain, and sidechain-to-water interactions. Nuclear magnetic resonance measurements additionally show that portions of disordered residues 151-214 remain highly dynamic in FUS-LC-C fibrils and that fibrils formed by the N-terminal half of the FUS LC domain (FUS-LC-N, residues 2-108) have the same core structure as fibrils formed by the full-length LC domain. These results contribute to our understanding of the molecular structural basis for amyloid formation by FUS and by LC domains in general.

[1] Laboratory of Chemical Physics, National Institute of Diabetes and Digestive and Kidney Diseases, National Institutes of Health, Bethesda, Maryland 20892-0520, USA. [2] Department of Biochemistry, University of Texas Southwestern Medical Center, 5323 Harry Hines Boulevard, Dallas, Texas 75390-9152, USA. [3] Institute for Quantum Life Science, National Institutes for Quantum and Radiological Science and Technology, Chiba 263-8555, Japan. ✉email: robertty@mail.nih.gov

Current interest in self-assembly processes involving proteins with domains of low sequence complexity (LC domains) arises from observations that liquid–liquid phase separation (LLPS) by such proteins within eukaryotic cells has a variety of biological functions[1,2], that LLPS by proteins with LC domains can lead to hydrogel formation in which their LC domains form the structurally ordered cores of amyloid-like fibrils[3,4], and that amyloid aggregation by proteins with LC domains is associated with neurodegenerative diseases[5–7].

FUS (FUsed in Sarcoma) is a 54 kDa RNA-binding protein with multiple functions, including roles in the regulation of transcription through interactions with the C-terminal domain of RNA polymerase II[8,9], in RNA splicing through multimeric binding to long introns[10,11], in DNA damage repair[12,13], in cellular responses to stress[14,15], and in mRNA transport and subcellular localization of translation[16]. The C-terminal half of FUS contains RNA recognition, zinc finger, Arg-Gly-Gly, and nuclear localization signal (NLS) domains. The N-terminal half is an LC domain comprised primarily of Ser, Gly, Gln, Tyr, Thr, and Pro residues, and with no Leu, Ile, Val, or Phe residues. Although the LC domain of FUS is largely unstructured as a monomer in solution[17], the LC domain is essential for normal FUS functions, including shuttling between the nucleus and cytoplasm[18], DNA damage repair[12], and transcription regulation[19,20].

In vitro, solutions of either full-length FUS or its LC domain alone undergo LLPS in a temperature-dependent and concentration-dependent manner. Phase-separated high-concentration droplets of full-length FUS or its LC domain typically evolve over time to a hydrogel state, consisting of highly entangled, amyloid-like fibrils[3], or to precipitated fibril bundles[21,22]. Thus, liquid-like FUS droplets represent a metastable condition[15], with the thermodynamic end state being FUS fibrils.

LC-domain-dependent intracellular aggregation of FUS[23,24] is also implicated in neurodegenerative diseases, especially familial amyotrophic lateral sclerosis and frontotemporal dementia, with disease-associated mutations occurring in both the NLS and the LC domains[25–27]. These mutations are believed to stabilize the fibrillar state of FUS[28] and promote its cytoplasmic localization[24], leading to either loss-of-normal-function[8,10,23] or gain-of-toxic-function[11,27,29] mechanisms for neurotoxicity.

Motivated by these considerations, our laboratory recently used solid-state nuclear magnetic resonance (ssNMR) measurements to determine the molecular structure of amyloid-like fibrils formed by the full-length LC domain of FUS (FUS-LC, residues 2–214). Unexpectedly, we found that a specific 57-residue segment in the N-terminal half of FUS-LC (underlined in Fig. 1a) forms the structurally ordered cross-β-fibril core, whereas the C-terminal half of FUS-LC remains dynamically disordered[21,30,31]. Eight different algorithms for predicting amyloid-forming segments from protein sequences failed to reproduce the experimental result[21], highlighting the qualitative difference between FUS-LC and amyloid-forming polypeptides such as amyloid-β, α-synuclein, and tau, whose self-assembly is driven chiefly by hydrophobic interactions.

Here we report studies of the C-terminal half of FUS-LC (FUS-LC-C, residues 111–214). Although this segment is dynamically disordered in fibrils formed by full-length FUS-LC, we find that FUS-LC-C by itself readily forms highly ordered fibrils, suitable for molecular structure determination by cryogenic electron microscopy (cryo-EM). The cryo-EM density map (2.62 Å resolution) and the resulting molecular model[32] show that the FUS-LC-C fibril core is qualitatively different from the core of full-length FUS-LC fibrils, in terms of both molecular conformation and overall symmetry. All-atom molecular dynamics (MD) simulations based on the cryo-EM structure provide insights into the properties of internal water and the nature of

sidechain–sidechain and sidechain–backbone interactions that contribute to FUS-LC-C fibril stability.

Although we do not claim that the FUS-LC-C fibril structure discussed below has direct relevance to pathogenesis or biological function, this work provides new information about the diversity of structural motifs that can exist within LC protein fibrils, their relation to motifs in fibrils formed by other types of sequences, and the diversity of intermolecular and inter-residue interactions that can stabilize amyloid fibril structures. The majority of disease-associated mutations in the LC domain of FUS do occur after residue 111[27].

## Results

### Both the N-terminal half and the C-terminal half FUS-LC form amyloid-like fibrils.
In fibrils formed by full-length FUS-LC with the sequence in Fig. 1a, residues 39–95 form the structurally ordered cross-β core, whereas the C-terminal half of the sequence is mainly dynamically disordered[21,30]. However, when FUS-LC is truncated to residues 111–214, the resulting FUS-LC-C construct readily forms amyloid-like fibrils under common solution conditions (40 μM protein concentration, 20 mM Tris-HCl buffer pH 7.4, 24 °C). In negative-stain transmission electron microscope (TEM) images, FUS-LC-C fibrils have a pronounced apparent width modulation with a period of 44 ± 4 nm (Fig. 1b). When FUS-LC is truncated to residues 2–108, the resulting N-terminal construct (FUS-LC-N) also forms fibrils, as expected (Fig. 1c).

Two-dimensional (2D) $^{15}$N-$^{13}$C ssNMR spectra of FUS-LC-C fibrils show sharp cross-peak signals, with full-width at half-maximum linewidths of 1.2 and 0.6 p.p.m. in $^{15}$N and $^{13}$C dimensions, respectively, consistent with a well-ordered structure. However, the extreme LC character of the FUS-LC-C sequence (34.6% Gly, 22.1% Ser, 19.2% Gln, 10.6% Tyr, 12.5% other) leads to 2D spectra with clusters of overlapping cross-peaks (Fig. 1d and Supplementary Fig. 1a). The 2D spectra of FUS-LC-N fibrils also show overlapping cross-peaks, although with more cross-peaks that are fully or partially resolved. Cross-peak positions in 2D spectra of FUS-LC-N fibrils are in good agreement with cross-peak positions in 2D spectra of full-length FUS-LC fibrils (Fig. 1e and Supplementary Fig. 1b). Specifically, 35 resolved or partially resolved cross-peaks in the 2D $^{15}$N-$^{13}$C spectrum of FUS-LC-N fibrils have chemical shift values that agree to within the ssNMR linewidths with those of cross-peaks in the corresponding spectrum of FUS-LC fibrils (Fig. 1e). More than 50 resolved or partially resolved cross-peaks in the 2D $^{13}$C-$^{13}$C spectrum of FUS-LC-N fibrils have chemical shift values that agree to within the ssNMR linewidths with those of cross-peaks in the corresponding spectrum of FUS-LC fibrils (Supplementary Fig. S1b). Thus, the molecular structure of the FUS-LC-N fibril core is very similar to the core structure in full-length FUS-LC fibrils determined earlier[21].

Measurements of the mass-per-length (MPL) of FUS-LC-C fibrils by dark-field TEM[33] yielded a value of 41.8 ± 1.0 kDa/nm (Supplementary Fig. 2a–c). Given that the molecular weight of a FUS-LC-C monomer is 10.02 kDa and the spacing between β-strands in a β-sheet is 0.47–0.48 nm, the MPL value for a single cross-β subunit in FUS-LC-C fibrils is expected to be about 21 kDa/nm, assuming a cross-β motif with one monomer per β-sheet spacing. The experimental MPL value therefore suggests a structure containing two cross-β subunits. In contrast, full-length FUS-LC fibrils contain a single cross-β unit[21].

### Cryo-EM reveals a two-fold symmetric core in FUS-LC-C fibrils.
To determine the molecular structure of FUS-LC-C fibrils, we used cryo-EM and helical reconstruction in RELION 3.0[34,35], with software modifications to include orientational correlations

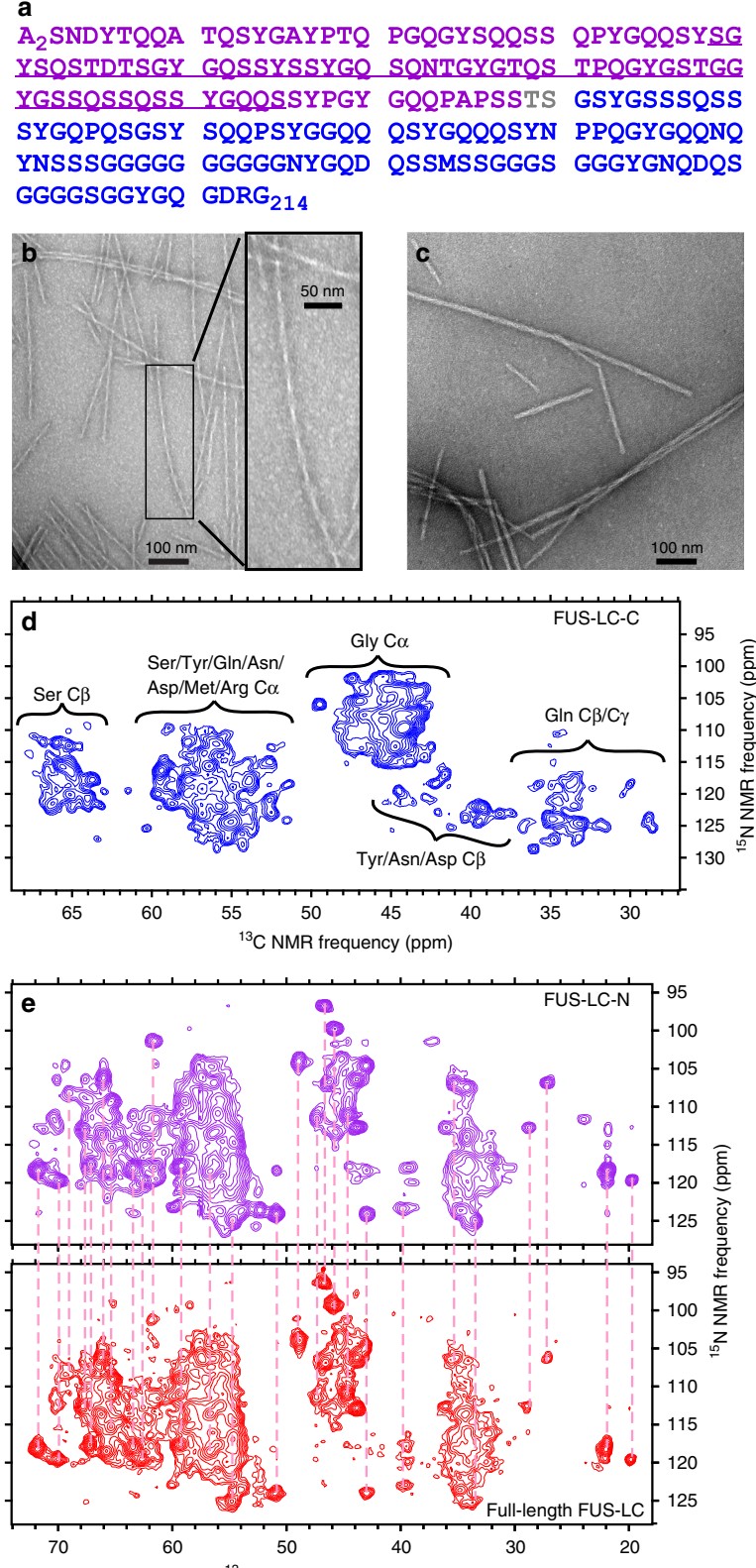

about the fibril growth direction for particles extracted from the same fibril segment[36]. Conditions are given in Table 1. Morphologically uniform, rapidly twisting fibrils were observed in cryo-EM images (Fig. 2a). β-Strands with the expected spacing of about 4.8 Å were clearly observed in 2D class images (Fig. 2b). Three-dimensional (3D) helical reconstruction followed by 3D

refinement with pseudo-$2_1$ symmetry yielded a final density map with 2.62 Å average resolution[32], based on the standard Fourier shell correlation criterion (Fig. 2c and Supplementary Fig. 3). In calculations without symmetry in the helical repeat unit, the final density map was nearly the same but with 2.70 Å resolution (Supplementary Fig. 3). Calculations in which $C_2$ symmetry was

**Fig. 1 Fibril formation by full-length FUS-LC and by its N-terminal and the C-terminal halves. a** Full-length FUS-LC sequence, residues 2–214. Segments in purple and blue lettering are FUS-LC-N and FUS-LC-C, respectively. The segment that forms the structurally ordered core of full-length FUS-LC fibrils is underlined. **b** Negative-stain TEM image of FUS-LC-C fibrils. Inset shows the rapidly twisting morphology. The same fibril morphology was observed in 53 images. **c** Negative-stain TEM image of FUS-LC-N fibrils. The same fibril morphologies were observed in 20 images. **d** 2D NCACX ssNMR spectra of FUS-LC-C fibrils, with residue-type assignments of cross-peak clusters. **e** 2D NCACX ssNMR spectra of uniformly $^{15}$N,$^{13}$C-labeled FUS-LC-N fibrils (purple), and full-length FUS-LC fibrils (red). Vertical dashed lines, all with the same length, connect corresponding cross-peaks in the two spectra. All fibril samples were uniformly $^{15}$N,$^{13}$C-labeled. Contour levels in all 2D spectra increase by successive factors of 1.3 and were set to show approximately the same number of levels below the maximum signals in all spectra.

**Table 1 Cryo-EM data collection and density map reconstruction parameters.**

| Data collection | |
| --- | --- |
| Microscope | Titan Krios |
| Camera | Gatan K2 Summit |
| Voltage | 300 kV |
| Magnification | ×130,000 |
| Defocus range | −1.0 μm to −2.5 μm |
| Pixel size | 1.07 Å |
| Exposure time (s) | 6 s |
| Electron dose | 47 e−/Å$^2$ |
| Reconstruction, EMDB code EMD-22169 | |
| Box size | 400 pixels |
| Interbox distance | 35.89 Å |
| Number of initial particles | 499,206 |
| Number of final particles in 3D reconstruction | 275,520 |
| Symmetry imposed | Pseudo-2$_1$ |
| Helical rise/helical twist | 2.44 Å/178.94° |
| Map resolution | 2.62 Å |
| Map sharpening B-factor | −83.2 Å$^2$ |

imposed resulted in a density map with a similar cross-section, but with unphysical connections between neighboring repeats along the fibril growth direction.

Helical rise and twist parameters for the density map in Fig. 2c are 2.44 Å and 178.94°, repsectively. The left-handed twist of the density map is consistent with atomic force microscope (AFM) height images (Supplementary Fig. 2d), which show left-handed asymmetry at crossovers. The 414.3 Å crossover period of the density map is in good agreement with the average crossover periods of 432 Å and 440 Å in AFM and TEM images, respectively (Supplementary Fig. 2e).

Assuming that the repeat unit is one complete FUS-LC-C molecule (including both ordered and disordered segments), the density map implies MPL = 41.1 kDa/nm, in good agreement with the dark-field TEM result. Cross-sectional views of the density map show that, within each cross-β subunit (purple and cyan in Fig. 2c), structurally ordered portions of FUS-LC-C molecules adopt U-shaped conformations, with two extended segments connected by a sharp bend or loop. Each cross-β subunit contains two layers of in-register parallel β-sheets.

**Molecular model for the FUS-LC-C fibril core**. The density map from cryo-EM shows that >50% of the residues in the structurally ordered core have bulky side chains, and that these residues occur sequentially, implying that the Ser/Gly-rich segments of FUS-LC-C (residues 163–175, 182–193, and 199–207) are not in the fibril core. We therefore focused on residues 111–160 in our attempts to fit the FUS-LC-C sequence into the cryo-EM density map for the fibril core. The clearly resolved side-chain densities could only be fit with a model comprised of residues 112–150, with residues 112–127 and 132–150 forming the outer and inner extended segments, respectively (Fig. 3a). This model accounts for nearly

all of the high-density regions of the map (Supplementary Fig. 4) and is consistent with dynamical information from NMR measurements described below. In contrast, models that comprised other segments of FUS-LC-C resulted in poor agreement with the density map, with multiple empty side-chain densities and side chains of the molecular models extending outside of the density (Supplementary Fig. 5).

Corrugations of backbone density align with the directions of backbone carbonyl groups in β-strand segments (Fig. 3b). If we replace the left-handed density map with its right-handed mirror image, backbone density corrugations no longer align with backbone carbonyl group directions in the molecular model that fits the mirror-image density (Supplementary Fig. 6). Thus, both molecular modeling and AFM images (Supplementary Fig. 2d) support a left-handed twist for FUS-LC-C fibrils.

The molecular model in Fig. 3a was created initially in Coot[37], then refined by simulated annealing in Xplor-NIH[38], using the density map as a restraint on atomic positions (probDistPot potential in Xplor-NIH). Conditions are given in Table 2. Structure bundles generated by multiple independent Xplor-NIH calculations[32] show that the molecular conformation is defined more precisely than the nominal resolution of the density map would indicate (Supplementary Fig. 7a–c). In structure bundles calculated with or without an additional statistical potential to restrain side-chain conformations (TorsionDB potential in Xplor-NIH), the root-mean-squared-displacement (RMSD) values for all backbone atoms of residues 113–149 are 0.30 Å and 0.26 Å, respectively. RMSD values for all non-hydrogen atoms of residues 113–149 are 0.79 Å and 0.75 Å, respectively. The fact that many side-chain conformations are uniquely determined, so that the all-atom RMSD is substantially less than 2.63 Å, is attributable of the asymmetric shapes of the side-chain densities (Supplementary Fig. 7d).

The molecular model shows that residues 113–122, 135–136, and 139–149 form the β-strands of in-register parallel β-sheets. Here, β-strands are defined to be segments in which backbone amide and carbonyl groups form intermolecular hydrogen bonds along the fibril growth direction and successive amino acid side chains are on alternating faces of the β-sheets. Residues 123–134 adopt an irregular conformation that allows the protein chain to fold back on itself. Non-β-strand conformations at G137 and G138 break residues 135–149 into two discontinuous β-strands, although this segment remains fully extended.

Side chains of most Gln residues in residues 112–150 are buried within the FUS-LC-C fibril core, with the sole exception of Q124. In contrast, the majority of Ser side chains are exposed on the surface of the structurally ordered core, interacting with solvent or possibly with disordered segments in residues 151–214. Thus, burial of Gln side chains appears to be a factor that determines the core structure. The larger number of glutamines in residues 111–150 (11), compared with residues 151–214 (9), may contribute to the thermodynamic preference for residues 112–150 as the core-forming segment of FUS-LC-C. The smaller number of glycines in residues 111–150 (7 vs. 29) is also expected to be an important factor.

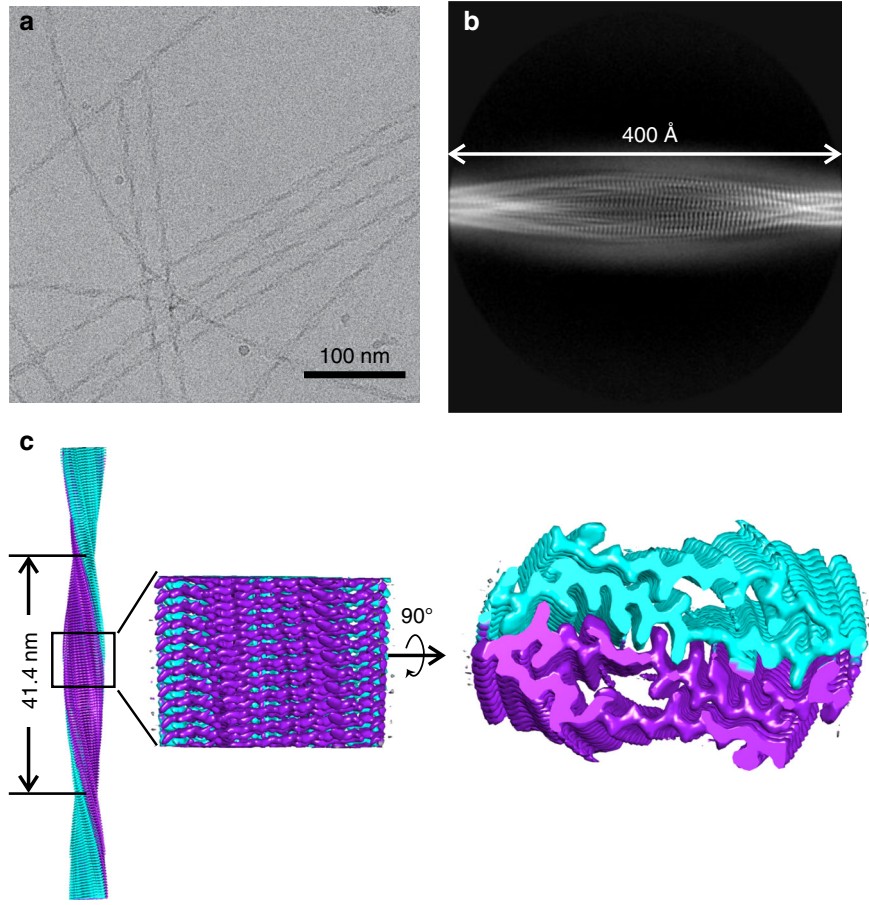

**Fig. 2 Cryo-EM density map of FUS-LC-C fibrils. a** Representative cryo-EM image of FUS-LC-C fibrils. **b** Example of a 2D class average image, nearly spanning one crossover period. **c** 3D density map reconstruction from 275,520 particles. Nominal resolution is 2.63 Å. The density has a left-handed twist and is reconstructed with quasi-$2_1$ symmetry, defined by a helical rise of 2.44 Å and twist of 178.94°. Densities for the two cross-β subunits are colored cyan and purple.

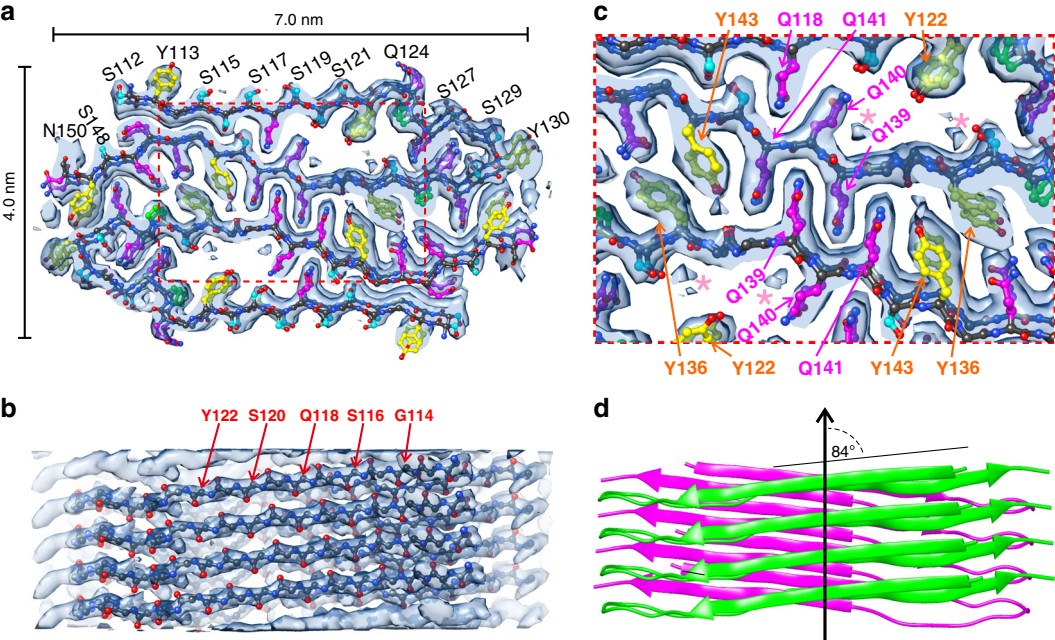

**Fig. 3 Molecular structural model for the FUS-LC-C fibril core, consisting of residues 112–140. a** Cross-sectional view, with side-chain carbon atoms of Ser, Tyr, Gln, and Pro residues colored cyan, yellow, magenta, and green, respectively. **b** Side view, showing that the directions of backbone carbonyl groups in the molecular model align with backbone corrugations in the density map, consistent with the left-handed twist of the density map. **c** Expanded view of the interface between cross-β subunits. Pink asterisks indicate density attributable to ordered water molecules. **d** Cartoon representation of the molecular model, illustrating the 12° angle between β-strands of the two cross-β subunits (green and magenta) and 84° angle to the fibril growth direction (black arrow).

**Table 2 Xplor-NIH potentials and structure validation statistics.**

| Xplor-NIH potentials | |
| --- | --- |
| Potential | Scale factor |
| probDistPot (cryo-EM density map) | 5.0 |
| posDiffPot (2.5 Å bound, backbone atoms only) | 10,000 |
| NCS (conformational equivalence) | 100 |
| DistSymmpot (translational symmetry) | 100 |
| TorsionDB | 0.002–2.0 |
| HBDB | Default |
| RepelPot | 0.004–4.0 |
| BOND | Default |
| ANGL | 0.4–1.0 |
| IMPR | 0.1–1.0 |
| Structure validation statistics, PDB code 6XFM | |
| MolProbity clash score | 2.0 |
| Ramachandran favored | 93.0% |
| Ramachandran allowed | 7.0% |
| Ramachandran outliers | 0.0% |
| RMSD of bonds | 0.01 Å |
| RMSD of angles | 1.50° |
| Number of non-hydrogen atoms | 2360 |
| Number of residues | 39 |

An expanded view of the central region of the fibril core (Supplementary Fig. 3) shows a classic steric zipper interdigitation of Gln side chains[39], both within each cross-β subunit (Q118 and Q140) and in the interface between subunits (Q139 and Q141). Amide groups of Gln side chains can form linear hydrogen-bonded chains, i.e., polar zippers[40], which stabilize the in-register, parallel cross-β organization (together with backbone hydrogen bonds). The interface between subunits also includes contacts between aromatic side chains of Y136 and Y143, possibly creating favorable π-stacking interactions[41]. Along the fibril growth direction, π-stacking interactions of Tyr side chains may also contribute to the stability of the in-register, parallel cross-β structure[42]. Additional aspects of interactions that stabilize the FUS-LC-C core structure are discussed below.

The angle between β-strands and the fibril growth direction is about 84° (Fig. 3d). This angle, together with the quasi-$2_1$ symmetry, creates a complex pattern of contacts between the two cross-β subunits. For example, if the Q139 side chain of molecule $j$ in the first subunit is surrounded by Q139 and Q141 side chains of molecules $k$ and $k+1$ in the second subunit, then the Q145 side chain of molecule $j$ is near Y136 side chains of molecules $k+1$ and $k+2$, whereas the Q132 side chain of molecule $j$ is near Y149 side chains of molecules $k$ and $k-1$. Thus, each molecule in one subunit interacts with four molecules in the other subunit. The complexity of these inter-subunit interactions contributes to the stability of the FUS-LC-C core structure in the MD simulations discussed below and presumably affects the kinetics and mechanism of fibril growth.

**The C-terminal segment of FUS-LC-C is dynamically disordered.** To identify dynamically disordered residues, we acquired one-dimensional (1D) $^{13}$C and 2D $^1$H-$^{13}$C NMR spectra of uniformly $^{15}$N,$^{13}$C-labeled FUS-LC-C fibrils at 24 °C using the Insensitive Nuclei Enhanced by Polarization Transfer (INEPT) technique for scalar coupling-driven $^1$H-$^{13}$C polarization transfers[43]. Signals that appear in the INEPT spectra can only arise from segments of FUS-LC-C that execute large-amplitude motions on sub-microsecond timescales[21,44,45]. The 1D and 2D INEPT spectra (Fig. 4a, b) show strong signals near random-coil $^1$H and $^{13}$C chemical shifts of Ser, Tyr, Gln, Asn, Asp, Met, and

Arg residues[46]. Signals with chemical shifts of Pro residues are weak. The presence of Arg and Asp signals, and the weakness of Pro signals indicate that the dynamically disordered segment begins after residue 150, consistent with the molecular model developed above.

For spectra in Fig. 4, the INEPT pulse sequence has the form $90_{X,H}-\tau_1-180_{X,H}180_{X,C}-\tau_1-90_{Y,H}90_{Y,C}-\tau_2-180_{X,H}180_{X,C}-\tau_2$. The maximum peak area for the Gly $C_\alpha$ signal spectra occurs at $\tau_{1,max} \approx 1.4$ ms and $\tau_{2,max} \approx 0.7$ ms, whereas the other $C_\alpha$ peaks have their maximum total area at $\tau_{1,max} \approx 1.2$ ms and $\tau_{2,max} \approx 1.2$ ms (Fig. 4c). Assuming transverse spin relaxation times $T_{2H}$ and $T_{2C}$ for $^1H_\alpha$ and $^{13}C_\alpha$ nuclei, the peak areas are expected to be proportional to $\sin(2\pi J \tau_1) \sin(2\pi k J \tau_2) \exp\left(-\frac{2\tau_1}{T_{2H}}\right) \exp\left(-\frac{2\tau_2}{T_{2C}}\right)$, where $J \approx 140$ Hz is the $^1H_\alpha$-$^{13}C_\alpha$ scalar coupling constant, with $k$ being the number of $^1H_\alpha$ nuclei (i.e., $k = 2$ for Gly $C_\alpha$; $k = 1$ for other $C_\alpha$ sites). The relaxation times are then $T_{2H} = \tan(2\pi J \tau_{1,max})/(\pi J)$ and $T_{2C} = \tan(2\pi k J \tau_{2,max})/(k\pi J)$. From these expressions, we estimate that the average relaxation times for Gly residues are $T_{2H} \approx 6.4$ ms and $T_{2C} \approx 3.2$ ms, and transverse spin relaxation reduces the maximum $^{13}C_\alpha$ signal of Gly residues in the 1D INEPT spectra by a factor of 0.37 relative to the ideal maximum value. For other residues, $T_{2H} \approx T_{2C} \approx 4.0$ ms, and transverse spin relaxation reduces their maximum $^{13}C_\alpha$ signals by a factor of 0.23. Figure 4c shows that the ratio of maximum 1D INEPT signals for Gly and non-Gly residues is 0.42. Taking into account the signal reduction factors, the ratio of dynamically disordered Gly residues in FUS-LC-C fibrils to dynamically disordered non-Gly residues is approximately $(0.23/0.37) \times 0.42 = 0.26$. In other words, Gly residues represent about $0.26/(1 + 0.26) = 21\%$ of the dynamically disordered residues. As Gly residues constitute 45% of residues 151–214 (Fig. 1a), only a subset of these disordered residues execute motions with sufficiently large amplitude on sufficiently short timescales to contribute to the INEPT spectra.

Thirteen Ser residues are contained within the core-forming segment of FUS-LC-C, in rough agreement with the number of resolved or partially resolved Ser $C_\beta$ cross-peaks in Fig. 1d. However, only six Gly residues are contained within the core-forming segment, whereas the 2D $^{15}$N-$^{13}$C ssNMR spectrum shows signals from at least 14 inequivalent Gly residues. This observation is consistent with the conclusion from INEPT spectra that only a subset of the Gly residues are dynamically disordered.

**Internal water and stabilizing interactions revealed by MD simulations.** Each cross-β subunit of the FUS-LC-C fibril core contains a pore defined by side chains of Y122, S135, and Q140 and by Gly residues 137 and 138 (Fig. 3a). To investigate the properties of water within this pore, to identify other locations for internal water molecules, and to gain additional insights into interactions that stabilize the FUS-LC-C core structure, we performed all-atom MD simulations on five repeats of the $2_1$-symmetric structure (i.e., ten copies of FUS-LC-C residues 112–150) in explicit solvent at 303 K. Two trajectories were analyzed from calculations with NAMD software[47], one for 400 ns in which position constraints were applied to all $C_\alpha$ sites, to ensure that the core structure did not disassemble during the simulation, the other for 500 ns without position constraints (see Supplementary Movies 1 and 2). With constraints, the backbone RMSD of the entire protein assembly was ~0.8 Å relative to the initial configuration throughout the 400 ns trajectory. Even without constraints, the core structure remained intact over the 500 ns trajectory, with the backbone RMSD of the six central chains remaining below 2.3 Å relative to the initial configuration (Supplementary Fig. 8a). We attribute the resistance of the core

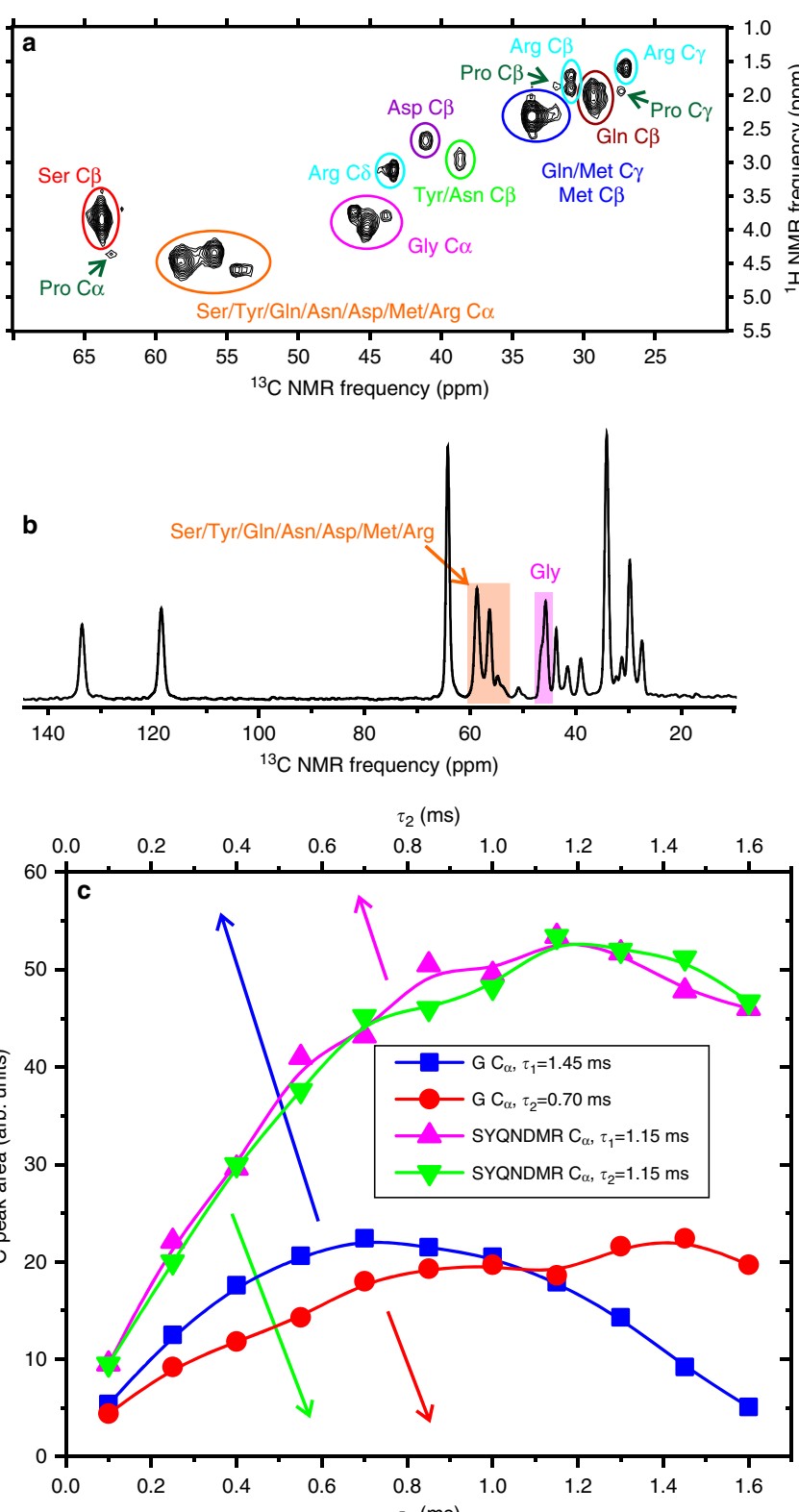

**Fig. 4 Detection of dynamically disordered segments by NMR. a** Aliphatic region of the 2D $^1$H-$^{13}$C INEPT spectrum of uniformly $^{15}$N,$^{13}$C-labeled FUS-LC-C fibrils. Residue-type assignments of cross-peaks are based on the known random-coil $^1$H and $^{13}$C chemical shifts of each residue type. **b** 1D $^{13}$C INEPT spectrum. $C_\alpha$ signals of Gly residues (pink) and other residues (orange) are highlighted. **c** Dependences of $C_\alpha$ peak areas in 1D INEPT $^{13}$C spectra on polarization transfer periods $\tau_1$ and $\tau_2$, for Gly residues (red and blue symbols, respectively) and other residues (green and pink symbols, respectively). Solid lines are guides to the eye.

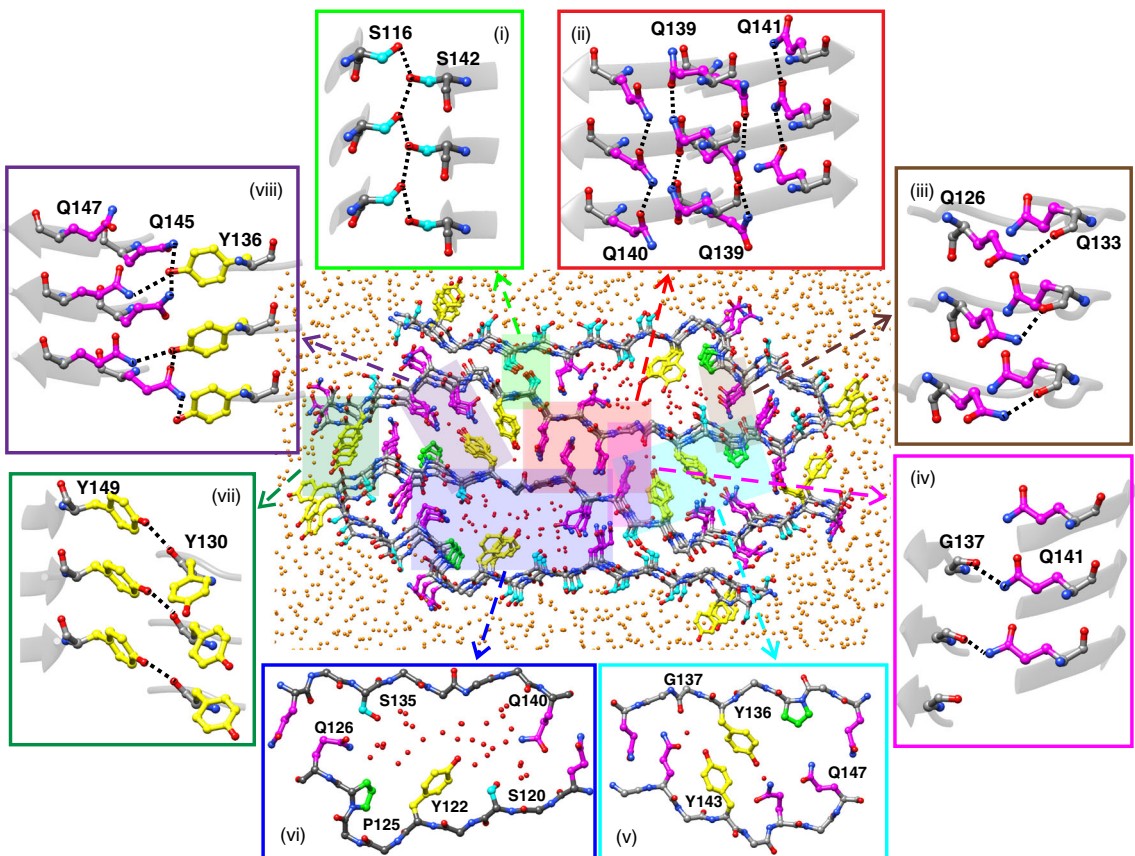

**Fig. 5 Internal water and side-chain interactions in the FUS-LC-C fibril core, from all-atom molecular dynamics simulations in explicit solvent.** A cross-sectional slab that includes three repeats of the cryo-EM-based structure is shown after 400 ns of simulated dynamics, with position constraints on backbone C$_\alpha$ atoms. Side-chain carbon atoms of Ser, Tyr, Pro, and Gln or Asn residues are shown in cyan, yellow, green, and magenta, respectively. Oxygen atoms of internal and external water molecules are shown in red and orange, respectively. Significant features include (i) S116–S142 side-chain hydrogen-bonding chains along the fibril growth direction, (ii) intermolecular Gln–Gln polar zipper interactions along the growth direction, (iii) hydrogen bonds between Q126 side-chain amide and Q133 backbone carbonyl groups, (iv) hydrogen bonds between Q141 side-chain amide and G137 backbone carbonyl groups, (v) isolated internal water molecules, (vi) water within internal pores, lined with side chains of Ser, Tyr, and Gln residues, (vii) hydrogen bonds between Y149 side-chain hydroxyl and Y130 backbone carbonyl groups, and (viii) hydrogen bond networks involving side chains of Y136, Q147, and Q145.

structure to structural distortions, despite the fact that 40% of FUS-LC-C molecules in these MD simulations were located at the fibril ends, to the complex pattern of inter-subunit interactions discussed above.

In the constrained MD simulation, residues 113–122, 135–136, and 139–149 maintained their β-strand character throughout the trajectory. In the unconstrained MD simulation, fraying occurred at the N- and C-terminal ends, so that the stable β-strand segments became residues 115–122, 135–136, and 139–147. Left-handed helicity persisted throughout the unconstrained simulation, with a final value of approximately −2.5° per 4.82 Å repeat.

As shown in Fig. 5, water molecules filled the main pores of the two cross-β subunits (Fig. 5, vi). Isolated water molecules also entered the core at other locations, especially near hydroxyl and amide groups of Tyr and Gln side chains (Fig. 5, v). The density of internal water equilibrated within 0.7 ns in the unconstrained trajectory (Supplementary Fig. 8b). All internal water molecules were mobile on the nanosecond time scale (Supplementary Fig. 8c), but average times for internal water molecules to move 10 Å were at least a factor of 10 greater than for external, bulk water (Supplementary Fig. 8d). Occupancy of hydrogen bonds to internal water was greatest for side-chain hydroxyl and amide groups around the main pores (S120, Y122, S135, and Q140), but was also relatively large for Q118, Q126, Y136, and Y143 (Supplementary Fig. 9a). Density attributable to partially ordered

water molecules is observed near side chains of Q140 and S135 in the cryo-EM density map (Fig. 3c), consistent with the MD simulations.

Previous studies based on crystallography[48,49] and ssNMR[30] have identified side-chain hydrogen bonds as likely stabilizing interactions for cross-β assemblies formed by LC sequences. Several distinct types of side-chain hydrogen bonds were apparent in our MD simulations. Most surprising was the observation of highly occupied sidechain-to-backbone hydrogen bonds between β-sheet layers (Fig. 5, iii, iv, and vii), including hydrogen bonds between side-chain amide groups of Q126 or Q141 and backbone carbonyl groups of Q133 or G137, respectively, and side-chain hydroxyl groups of Y149 and backbone carbonyl groups of Y130. Sidechain-to-backbone hydrogen bonds were found both within each cross-β subunit (Supplementary Fig. 9b) and between subunits (Supplementary Fig. 9c). Sidechain-to-sidechain hydrogen bonds between β-sheet layers were also observed (Fig. 5, i and viii), including intra-subunit hydrogen bonds beween hydroxyl groups of S116 and S142 (Supplementary Fig. 9b) and inter-subunit hydrogen bonds between hydroxyl groups of Y136 and amide groups of Q145 and Q147 (Supplementary Fig. 9c).

Side-chain hydrogen bonds were also observed along the the fibril growth direction (Fig. 5, i and ii). In addition to the expected polar zipper interactions among Gln side chains (Supplementary

Fig. 9d) and among N150 side chains, these included intermolecular hydrogen bonds between side-chain hydroxyl and backbone amide groups of S131, and the aforementioned S116–S142 hydrogen bonds (Supplementary Fig. 9e).

Finally, the MD simulations revealed a diverse spectrum of side-chain dynamics. All six Tyr residues were found to have strongly preferred side-chain conformations that presumably optimize π-stacking interactions along the fibril growth direction[42], corresponding to side-chain torsion angles $\chi_1 \approx -65°$ and $\chi_2 \approx \pm100°$. Tyr aromatic ring flips could then be visualized as rapid changes in the sign of $\chi_2$. In MD simulations with and without constraints, Y130, Y136, and Y143 did not undergo ring flips on the time scale of these simulations (Supplementary Figs. 10a and 11a, respectively). In contrast, in simulations without constraints, Y122 side chains, which project into the main water pores, did flip in the simulated trajectories, as did the solvent-exposed side chains of Y113 and Y149.

Orientations and dynamics of Gln side chains were quantified by means of a pseudo-torsion angle $\xi$, defined by the coordinates of the following four atoms: $N_{\varepsilon2}$, $O_{\varepsilon1}$, and $C_\alpha$ for a Gln residue in molecule $k$ of one cross-β subunit, and $C_\alpha$ for the same residue in molecule $k + 1$ of the same subunit. With this definition, if the $O_{\varepsilon1}$–$N_{\varepsilon2}$ direction (i.e., the direction of the side-chain amide group) is parallel or antiparallel to the fibril growth direction, $\xi$ is ~0° or approximately ±180°, respectively. In MD simulations without constraints (Supplementary Fig. 10b), side-chain amide groups of Q133, Q139, and Q141 maintained constant orientations throughout the 500 ns trajectory. These residues participated in stable polar zippers (Supplementary Fig. 9d). The Q126 side-chain orientation was also nearly constant, but with $\xi \approx 100°$, corresponding to stable intramolecular hydrogen bonds between Q126 side-chain amide and Q133 backbone carbonyl groups. The solvent-exposed side chain of Q124 exhibited rapid, nearly random orientational fluctuations. Other Gln side chains, including Q118, Q132, Q140, Q145, Q146, and Q147, exhibited transitions among preferred orientations, including orientations that allow polar zipper interactions ($\xi \approx 0°$, ±180°) and orientations that allow hydrogen bonds to nearby polar side chains, to internal water molecules, or to backbone carbonyl groups ($\xi \approx 50°$, ±100°). Similar behaviors were observed in MD simulations with backbone $C_\alpha$ restraints, indicating that the dynamics and interactions of Gln side chains do not depend on large-amplitude backbone motions or on large deviations of the backbone conformation from the initial cryo-EM-based molecular model.

## Discussion

### Comparison of FUS-LC-C and FUS-LC-N fibril structures.

According to ssNMR data in Fig. 1 and Supplementary Fig. 1, the core structure in FUS-LC-N fibrils is the same as in full-length FUS-LC fibrils, which was shown previously to contain an in-register, parallel cross-β stacking of residues 39–95 in an approximate S-shaped conformation[21,31]. Results described above show that the core structure in FUS-LC-C fibrils is qualitatively different, consisting of two in-register, parallel cross-β stacks of residues 112–150 in U-shaped conformations, with quasi-2₁ symmetry about the stacking direction (i.e., the fibril growth direction). Despite their structural differences, amino acid compositions of the two cores are similar (31.6% Ser, 21.1% Gly, 17.5% Gln, 14.0% Tyr, 10.5% Thr, 5.3% Asn/Asp/Pro in residues 39–95 of FUS-LC; 33.3% Ser, 28% Gln, 15.4% Gly, 15.4% Tyr, 7.7% Asn/Pro in residues 112–150 of FUS-LC). Potentially significant differences include the higher Gln content and the lower contents of Gly and hydroxyl-bearing residues in the FUS-LC-C core.

The FUS-LC-C core includes continuous β-strand segments with 10-residue and 11-residue lengths, with 59% of the core-forming sequence participating in β-sheets. In contrast, the two longest β-strand segments in the FUS-LC-N core are 5 and 7 residues in length, and 40% of the core-forming sequence participates in β-sheets. The term Low-complexity Aromatic-Rich Kinked Segments (LARKS) was introduced by Eisenberg and colleagues[48] to describe cross-β motifs in full-length FUS-LC and other LC-domain fibrils, and to distinguish these motifs from the wider or flatter β-sheets observed in other amyloid fibrils[50–54]. The FUS-LC-C core structure is not obviously LARKS-like. Rather, the β-sheet formed by residues 113–122 is quite flat. The β-sheet formed by residues 139–149 has a crease at G144, but side chains of these residues appear tightly packed and interdigitated, as in standard steric zipper structures[39].

### Factors that determine the structure of full-length FUS-LC fibrils.

The experimental fact that full-length FUS-LC fibrils have the FUS-LC-N core rather than the FUS-LC-C core requires explanation. One possibility is that a core formed by residues 39–95 is simply more stable than a core formed by residues 112–150. Measurements of the optical absorbance of supernatants from pelleted fibril suspensions (see "Methods" in Supplementary Information), indicate equilibrium solubilities of 4.5 ± 1.3 μM for FUS-LC-C and 2.7 ± 1.7 μM for FUS-LC-N under our experimental conditions (mean ± SD, three independent experiments). Solubility ratios of 1.9 ± 0.6 imply a difference in the free energy of fibril formation $\Delta\Delta G \approx RTln(1.9 \pm 0.6) = 0.35 \pm 0.19$ kcal/mol. Thus, FUS-LC-N fibrils are apparently more stable than FUS-LC-C fibrils under our experimental conditions, suggesting that the 39–95 core is indeed preferred thermodynamically. However, the difference is small. If the relative stability of the two cores were the only factor, one might expect two distinct FUS-LC fibril structures to coexist, as observed in the case of 40-residue amyloid-β (Aβ40) fibril polymorphs with similarly small differences in thermodynamic stability[55].

Kinetic factors may play a role. As described above, the FUS-LC-C core contains a complex pattern of contacts between the two cross-β subunits of the quasi-2₁-symmetric structure, with each molecule in one subunit interacting with four molecules of the other subunit. Formation of this structure may then be inherently slower than formation of the FUS-LC-N core structure, which consists of a single cross-β unit.

Another potentially important factor is the effect of fibril formation on protein segments outside the fibril core. As the FUS-LC-N and FUS-LC-C cores contain only 57 and 39 residues, respectively, more than 70% of the FUS-LC sequence remains disordered upon fibril formation. NMR spectra show that the disordered segments, or portions thereof, execute large-amplitude motions on sub-microsecond timescales, both in FUS-LC-C fibrils (Fig. 4) and in full-length FUS-LC fibrils[21]. However, the close proximity of the disordered segments of neighboring molecules within an in-register, parallel cross-β structure necessarily restricts the volume available to each disordered segment, leading to a substantial decrease in configurational entropy. A 100-residue disordered polypeptide in free solution adopts conformations characterized by a radius of gyration $r_g \approx 30$ Å[56]. If the available volume is limited to $15r_g^3$ when the same polypeptide is attached to a cross-β fibril core, the number of accessible conformations may decrease by a factor of 100[57], corresponding to a free energy contribution of $RTln(100) \approx 2.8$ kcal/mol.

The qualitative structural differences between the FUS-LC-N and FUS-LC-C cores may lead to substantial differences in the available volumes for disordered segments in the context of full-length FUS-LC. Specifically, the U-shaped conformation of residues 112–150 in the FUS-LC-C core would bring N-

terminal and C-terminal disordered segments in close proximity, whereas the S-shaped conformation of residues 39–95 in the FUS-LC-N core does not (Supplementary Fig. 12a). The greater entropy loss of disordered segments may therefore prevent formation of full-length FUS-LC fibrils with the FUS-LC-C core. In future experiments on other segments of FUS-LC, it may be possible to demonstrate the effects of disordered segments on the identities and structures of core-forming segments, using cryo-EM and ssNMR measurements. For example, in fibrils formed by residues 2-$X$ of FUS-LC, the core structure may switch from that of FUS-LC-N to that of FUS-LC-C as $X$ decreases from 214 to 150.

One might also ask why full-length FUS-LC fibrils that contain both core structures do not exist. A structure containing a central $2_1$-symmetric FUS-LC-C core and two flanking FUS-LC-N cores may seem plausible when viewed in cross-section (Supplementary Fig. 12b). However, the FUS-LC-C core has an inherent left-handed twist, rotating 360° in 88 ± 8 nm. The fact that variations in the twist period are small (Supplementary Fig. 2e) implies that untwisting the FUS-LC-N core would require work that greatly exceeds thermal energy. The centers of mass of FUS-LC-N cores in a putative triple-core structure must then follow spiral paths around the fibril growth direction (Supplementary Fig. 12c). With a twist period $L \approx 88$ nm and a radius $R \approx 5.5$ nm, the spiral path length would be $\sqrt{L^2 + 4\pi^2 R^2} \approx 95$ nm. Consequently, the average intermolecular spacing in the FUS-LC-N cross-β structure must increase by a factor of $95/88 = 1.08$, to 5.1–5.2 Å. The energetic cost of stretching hydrogen bonds or straining molecular conformations to accommodate the larger spacing may then preclude structures such as the one depicted in Fig. 5b.

Similar principles may affect the structure of amyloid fibrils formed by other proteins, especially fibrils formed by proteins larger than 15 kDa and proteins with uniform amino acid compositions. Entropy loss associated with segments outside the fibril core may determine the identities and conformations of core-forming segments. The diameter of the fibril core, and hence the length of the core-forming segment, may be limited by the energetics of intermolecular hydrogen bonds and molecular conformations at the perimeter of the core.

It should also be noted that FUS-LC-N contains 18 Gly and 7 Pro residues, whereas FUS-LC-C contains 36 Gly and 4 Pro residues. The larger population of Gly residues in FUS-LC-C may contribute to the identity of the core-forming segment in FUS-LC fibrils. The fact that Gly residues are concentrated after residue 165 contributes to the selection of residues 112–150 for the core of FUS-LC-C fibrils.

**Interactions within the FUS-LC-C fibril core**. MD simulations described above reveal several characteristics of the FUS-LC-C fibril core that are likely to be shared with other amyloid structures. In addition to water that fills the large internal pores around Y122 side chains, isolated water molecules occupy smaller cavities, especially around the side chains of Q132 and Y143. Residence times of water in large and small cavities are similar and roughly ten times longer than outside the core (Supplementary Fig. 8).

Hydrogen-bonding interactions within the core are surprisingly diverse. Polar zipper interactions within rows of Gln side chains vary in their occupancy and stability, being highly occupied and directionally stable for Q139, Q133, and Q141, but more variable for other Gln residues (Supplementary Figs. 9–11). Variations in dynamics and ordering of Gln side chains may explain the variability of Gln side-chain signals in ssNMR studies of amyloid fibrils, including full-length FUS-LC fibrils[21,30]. Similar considerations apply to Tyr side chains, which exhibit a range of ring flip rates (Supplementary Figs. 10 and 11).

In addition to polar zippers, Gln side chains can engage in hydrogen bonds to nearby backbone sites or polar side chains. For example, the Q126–Q133 interaction (Fig. 5, iii) apparently stabilizes the β-turn between these residues, together with hydrogen bonds between S131 hydroxyls and S129 carbonyls. Other highly occupied sidechain–backbone hydrogen bonds include Y149–Y130 and Q141–G137 interactions that contribute to the interface between cross-β subunits (Supplementary Fig. 9).

Direct sidechain–sidechain hydrogen bonds between different residues are less prevalent than sidechain–backbone hydrogen bonds. The most prominent example is the S116–S142 interaction, which apparently helps stabilize the interface between β-strand segments of a single subunit. Related examples of sidechain–backbone and sidechain–sidechain hydrogen bonds occur in cross-β crystal structures of amyloid-forming peptides[39,58], including peptides from FUS-LC[49,59]. Inter-subunit Ser-Ser sidechain–sidechain hydrogen bonds may exist in $2_1$-symmetric TDP43 fibrils[60]. ssNMR data indicate sidechain–backbone and sidechain–sidechain hydrogen bonds involving Ser and Thr residues in the core of full-length FUS-LC fibrils[30].

Hydrophobic interactions and salt bridges can also be important stabilizing interactions for pathogenic amyloid fibrils, as indicated by structural studies of amyloid-β[50,54,61,62] and α-synuclein[51,63] fibrils, as well as for functional amyloids[64–66]. The absence of charged amino acid side chains and the absence of purely hydrophobic residues other than Pro within the FUS-LC-C fibril core implies that hydrophobic interactions and salt bridges do not account for the formation of FUS-LC-C fibrils.

**Recurring structural motifs in amyloid fibrils**. U-shaped molecular conformations and two-fold symmetry about the growth direction were reported previously in early ssNMR-based structural models for synthetic Aβ40 fibrils[67], as well as in a recent cryo-EM structure of fibrils formed by residues 84–178 of human PrP[68]. Structural similarity to the FUS-LC-C fibril core reported above is especially remarkable in light of the qualitatively different amino acid compositions of Aβ40 and PrP (both of which contain multiple Ala, Phe, Val, Leu, Ile, and Met residues in their core-forming segments). As pointed out previously[21], the S-shaped molecular conformation in the FUS-LC-N core closely resembles the conformation in α-synuclein fibril cores[51]. A threefold symmetric core structure has been reported for Gln-rich Orb2 fibrils[52], similar to threefold-symmetric Aβ40 fibrils structures from ssNMR[61,69]. Apparently, recurring structural motifs in amyloid cores can be stabilized either by hydrophobic interactions or by a variety of side-chain hydrogen-bonding interactions, contributing to the nearly generic propensity of polypeptide chains to self-assemble into amyloid fibrils[70].

## Methods

**Production of FUS-LC-C and FUS-LC-N**. Recombinant FUS-LC-C (residues 111–214 in Fig. 1a) with an N-terminal green fluorescent protein (GFP) tag and FUS-LC-N (residues 2–108) with an N-terminal His$_6$ tag were expressed in BL21 (DE3) *Escherichia coli* cells. A linker segment ending with a caspase-3 cleavage site allowed tag-free FUS-LC-C to be produced by caspase-3 cleavage. A linker segment ending with a Tobacco Etch Virus (TEV) cleavage site allowed tag-free FUS-LC-N to be produced by TEV cleavage. Cells were grown in 1 L of Luria-Bertani medium (or M9 medium containing 2 g/L of $^{13}C_6$-glucose and 1 g/L of $^{15}N$-ammonium chloride for the ssNMR samples) containing 100 μg/mL ampicillin at 37 °C with shaking at 240 r.p.m. until the optical density (OD$_{600}$) reached 0.8–1.0. Protein expression was induced by adding 0.5 mM isopropyl β-D-1-thiogalactopyranoside and cells were further cultured at 20 °C overnight for GFP-FUS-LC-C or at 37 °C for 5 h for His$_6$-FUS-LC-N. After protein expression, the cells were centrifuged at 4000 × g for 30 min at 4 °C. The cell pellet was resuspended in 50 mL of lysis buffer (20 mM Tris-HCl pH 7.4, 500 mM NaCl, 0.5% v/v Triton X-100, 20 mM 2-mercaptoethanol, one tablet of SigmaFASTprotease inhibitor cocktail, 0.2 mg/mL

lysozyme, and either 2 M urea for GFP-FUS-LC-C or 2 M guanidine-HCl for His$_6$-FUS-LC-N) and sonicated in an ice bath for 10 min using a Branson Model 250 sonifier at 0.4 output, 30% duty cycle, with a tapered 1/8" microtip horn. The lysed cells were centrifuged at 140,000 × g for 1 h at 4 °C. The supernatant was loaded onto a gravity flow column containing 10 mL of nickel NTA agarose resin (Goldbio) and the column was placed in a rotator for 30 min at 4 °C to allow protein binding. The column was then washed with 250 mL of wash buffer (20 mM Tris-HCl pH 7.4, 200 mM NaCl, 20 mM 2-mercaptoethanol, 0.1 mM phe-nylmethylsulfonyl fluoride (PMSF), 25 mM imidazole, and either 2 M urea for GFP-FUS-LC-C or 2 M guanidine-HCl for His$_6$-FUS-LC-N) and eluted with the elution buffer (20 mM Tris-HCl pH 7.4, 200 mM NaCl, 20 mM 2-mercaptoethanol, 0.1 mM PMSF, 250 mM imidazole, and either 2 M urea for GFP-FUS-LC-C or 2 M guanidine-HCl for His$_6$-FUS-LC-N). Ethylenediaminetetraacetic acid (0.5 mM) was added to the eluted protein solutions.

Purified proteins were concentrated using an Amicon ultracentrifuge filter with a molecular weight cutoff of 10 kDa to ~50 mg/mL. GFP-FUS-LC-C was cleaved by diluting the protein to 2 mg/mL with a buffer consisting of 20 mM Tris-HCl pH 7.4, 200 mM NaCl, 20 mM 2-mercaptoethanol, and 0.1 mM PMSF, followed by addition of caspase-3 with a 1:1500 mass ratio of caspase-3 to GFP-FUS-LC-C. Similarly, His$_6$-FUS-LC-N was cleaved by diluting the protein to 2 mg/mL with a buffer consisting of 20 mM Tris-HCl pH 8.0, 150 mM NaCl, 1 mM dithiothreitol, followed by addition of TEV protease with a 1:200 mass ratio of TEV protease to His$_6$-FUS-LC-N. The resulting solution was rotated at room temperature overnight. Cleaved, tag-free proteins were then purified by fast protein liquid chromatography using a Superdex 200 PG column with 20 mM Tris-HCl pH 7.4, 200 mM NaCl, 20 mM 2-mercaptoethanol, 0.1 mM PMSF, and 2 M guanidine-HCl for 150 min at a 2 mL/min flow rate. FUS-LC-C or FUS-LC-N eluted at 115–125 min. Purified proteins were then concentrated to ~50 mg/mL in 6 M guanidine-HCl using an Amicon ultracentrifuge filter with a molecular weight cutoff of 3 kDa and stored at −80 °C.

Full-length FUS-LC, with the N-terminal His-tag MSYYHHHHHH-DYDIPTTENLYFQGAMDP, was expressed in BL21(DE3)PLysS E. coli cells at 37 °C. After cell lysis by sonication in 50 mM Tris-HCl pH 7.5 with 500 mM NaCl, 1% Triton X-100, 6 M guanidine-HCl, and protease inhibitor, lysed cells were pelleted at 223,000 × g and 4 °C. FUS-LC was purified from the supernatant by binding to a nickel NTA agarose column in the presence of 8 M urea, followed by elution with 20 mM sodium phosphate pH 7.4, 500 mM NaCl, 7.6 M urea, and 200 mM imidazole. Additional details were as previously described[21].

**Fibril formation.** An initial batch of FUS-LC-C fibrils was prepared by dialyzing the protein solution at 2 mg/ml in 6 M guanidine-HCl against 1 L of 20 mM Tris-HCl, 20 mM 2-mercaptoethanol, 0.1 mM PMSF at pH 7.4 overnight. The sample was then incubated at room temperature for 5–7 days. Fibril formation was verified by negative-stain TEM. Negative-stain TEM images showed that many of the FUS-LC-C fibrils occurred in multi-stranded bundles. To prepare single-stranded fibril seeds for a subsequent batch of fibrils, fibrils from the initial batch were pelleted at 280,000 × g for 1 h at 4 °C in a Beckman Optima ultracentrifuge. After this ultra-centrifugation step, single-strand fibrils were observed in the supernatant. These single-stranded fibrils were then sonicated with a Branson Model 250 sonifier and tapered 1/8" microtip horn at 0.1 output and 10% duty cycle for 10 min, yielding short fibril fragments (i.e., seeds). Fibril growth in the subsequent batch was then initiated by adding 5% seeds to 0.4 mg/mL of monomeric FUS-LC-C, which had been dialyzed against 1 L of 20 mM Tris-HCl, 20 mM 2-mercaptoethanol, 0.1 mM PMSF at pH 7.4 overnight. This solution was then incubated at room temperature for 4 days.

Approximately 20% of our cryo-EM images of FUS-LC-C fibrils contain minor fractions of non-fibrillar aggregates. As non-fibrillar aggregates are readily distinguished from fibrils in the manual particle picking process, they do not contribute to the density map or structural model for FUS-LC-C fibrils.

For NMR measurements, seeded fibrils, prepared as described above but using uniformly $^{15}$N,$^{13}$C-labeled FUS-LC-C, were pelleted at 280,000 × g and 4 °C for 1 h in a Beckman Optima ultracentrifuge. Fibril pellets were then packed into 3.2 mm magic-angle-spinning (MAS) rotors by centrifugation with a swinging-bucket rotor at 70,000 × g for 1 h at 4 °C, using a home-made funnel device to hold the MAS rotor and the centrifuge tube that contained the fibril pellet.

Uniformly $^{15}$N,$^{13}$C-labeled FUS-LC-N fibrils for ssNMR were prepared and packed with the same procedures.

**Measurements of FUS-LC-C and FUS-LC-N solubility.** FUS-LC-C and FUS-LC-N proteins were separately diluted to 3 mg/mL in 20 mM Tris-HCl pH 7.5, 200 mM NaCl, 20 mM 2-mercaptoethanol, 0.5 mM ETDA, 0.1 mM PMSF, and 6 M gua-nidine-HCl, and dialyzed overnight against 1 L of 20 mM Tris-HCl pH 7.5, 200 mM NaCl, 20 mM 2-mercaptoethanol, 0.5 mM ETDA, 0.1 mM PMSF, and 0.02% sodium azide. The protein solutions were transferred to tubes, the protein con-centrations were adjusted to 2 mg/ml with dialysis buffer, and the solutions were incubated at 24 °C for 1 week for full fibril formation. The resulting FUS-LC-C and FUS-LC-N fibrils were pelleted at 280,000 × g and 4 °C for 1 h. After the super-natants were removed, the fibril pellets were washed with dialysis buffer, then resuspended in fresh buffer by sonication. After incubation for three days to allow equilibration of fibrillar FUS-LC-C or FUS-LC-N with monomers in solution, the

fibrils were again pelleted at 280,000 × g and 4 °C for 1 h. Protein concentrations in the supernatants, representing equilibrium solubilities of FUS-LC-C or FUS-LC-N, were then determined from optical absorbance at 280 nm. The experiments were repeated three times.

**Transmission electron microscopy.** For negative-stain TEM, samples were pre-pared on grids consisting of a 3 nm carbon film supported by lacey carbon on 300-mesh copper (Electron Microscopy Sciences LC325-Cu-CC). For dark-field TEM, samples were prepared on grids consisting of 2 nm carbon films on 300-mesh copper (Quantifoil S 7/2). Grids were glow-discharged immediately before use. For negative-stain images, a 10 µL aliquot of fibril solution was applied to the grid. After 2 min adsorption to the carbon film, the grid was blotted and washed with 10 µL of water for 10 s, blotted, stained with 10 µL of 2% w/v uranyl acetate for 1 min, blotted, and dried in air. Negative-stain TEM images were obtained with a FEI Morgagni microscope, operating at 80 kV, using a side-mounted Advantage HR camera (Advanced Microscopy Techniques).

For dark-field images used in MPL measurements, 5 µL of fibril solution (0.2 mg/mL) and a 5 µL of tobacco mosaic virus (TMV) solution (0.46 mg/mL) were applied to a freshly glow-discharged grid. After 5 min of incubation, the grid was blotted, washed five times with 10 µL of water for 10 s each, blotted, and dried in air. Dark-field images were acquired with the same FEI Morgagni microscope, operating at 80 kV, using a bottom-mounted XR-550B camera (Advanced Microscopy Techniques), with an electron beam tilt angle of 1.2° and ×5600 magnification. Care was taken to ensure uniform illumination of the field-of-view by the electron beam, full blockage of the direct beam by the objective aperture of the microscope, and other conditions described previously[33].

Dark-field images were analyzed with ImageJ software (available at http://imagej.nih.gov/ij). Image intensities were integrated within 100 nm × 60 nm rectangular areas centered on TMV rods or FUS-LC-C fibrils ($I_{center}$), or in background regions on both sides of the TMV rods or FUS-LC-C fibrils ($I_{B1}$ and $I_{B2}$, see Supplementary Fig. 2a). Each MPL count was calculated according to the equation MPL = $(I_F/I_{TMV})$ × 131 kDa/nm, where $I_F = I_{center} - (I_{B1} + I_{B2})/2$ for one fibril segment and $I_{TMV}$ is the average value of $I_{center} - (I_{B1} + I_{B2})/2$ for many TMV rods. A total of 96 counts was used for the histogram in Supplementary Fig. 2b, with $I_{TMV}$ being the average of 52 measurements. For the MPL error distribution in Supplementary Fig. 2c, error values $E$ were determined from integrated background intensities $I_B$ according to the equation $E = \sqrt{\frac{3}{2}[(I_B - I_{B,ave})/I_{TMV}]}$ × 131 kDa/nm, where $I_{B,ave}$ is the average of the background intensities. Gaussian fitting of MPL and error histograms was performed with Igor Pro 8.03 (WaveMetrics).

**Atomic force microscopy.** Tapping mode AFM was performed on a Veeco Multimode AFM instrument with Nanoscope IV controller, using a PPP NCHAuD-10 probe (Nanosensors). For AFM measurements, a 2 µL aliquot of FUS-LC-C fibril solution (0.3 mg/mL) was combined with 18 µL of deionized water. The diluted fibril solution was applied to a freshly cleaved mica surface, which was then blotted, washed with 10 µL of water, blotted, and dried in air. AFM images were collected with a 3.12 Hz scan rate, 512 points/line, and 512 lines over a 1.0 µm × 1.0 µm area.

**Cryo-EM image acquisition.** Quantifoil R2/2 grids (SPI Supplies) with 300 mesh were glow-discharged immediately before use. A 2.5 µL aliquot of fibril solution (0.4 mg/mL) was applied to the grid with 10 s preblotting and 5 s blotting, then plunge-frozen into liquid ethane using a Leica EM GP2 plunger. Data were col-lected on a Titan Krios microscope, operating at 300 kV and using ×130,000 magnification. A total of 4425 micrographs was recorded with 1.07 Å pixel size, 47 electrons/Å$^2$ dose, 1.0–2.5 µm defocus range, and 6 s exposure time, using SerialEM 3.7 and AMT Image Capture Engine 6.02 software.

**Cryo-EM image processing.** From the 4425 micrographs, micrographs without fibrils and micrographs that contained defects or crystalline ice were manually deleted, leaving 2411 micrographs for image processing. All image processing steps were performed using RELION 3.0[34,35] and a modified version of RELION 3.0, with modifications to include correlations of orientations about the fibril growth direction (i.e., $\rho$ angles) for particles that came from the same fibril segment[36]. Single-stranded fibrils were manually selected and a total of 499,206 particles were extracted, using 400 pixel (428 Å) box size and an interbox distance of 35.89 Å (91.6% overlap). After 14 rounds of 2D classification, 113 2D classes with 406,815 particles were selected for 3D classification. Initial 3D classification was performed using a featureless cylinder as an initial model. We performed multiple rounds of 3D classification with the helical parameters estimated initially from the negative-stain TEM images of fibrils (i.e., 4.8 Å helical rise and −2.0° helical twist), then refined with the local helical symmetry search. After the initial round of 3D classification, we used in-house Matlab scripts to examine the consistency of angular alignments of particles and to fix in-plane rotation angles (i.e., $\psi$ angles) to make all particles from the same fibril segment have consistent rotational angles relative to the fibril growth direction. After three rounds of 3D classification with one class, a fourth round of 3D classification was performed with three classes, to include the possibility of polymorphism in the FUS-LC-C fibrils. The major class,

containing 69% of the particles (275,520 particles), yielded the best helical density map, with 3.79 Å resolution. The other two classes yielded lower-resolution density maps (6–7 Å resolution), which were not qualitatively different from the major class. Particles from the major class were used for further processing.

Three-dimensional auto-refinement and post-processing were then performed, improving the resolution to 3.11 Å. After contrast transfer function refinement and motion correction adjustments, 3D auto-refinement and post-processing were repeated. The resulting density map (Supplementary Fig. 3a) had 2.70 Å resolution, with helical rise and twist of 4.89 Å and −2.11°. Examination of this density map revealed nearly perfect $2_1$ symmetry. Therefore, 3D auto-refinement and post-processing were performed again, this time with quasi-$2_1$ symmetry, corresponding to initial values for helical rise and twist of $\sim 0.5 \times 4.89$ Å and $180° - 0.5 \times 2.11°$. The resolution of the final density map with quasi-$2_1$ symmetry (Figs. 2 and 3) was 2.62 Å, and helical rise and twist of 2.44 Å and 178.94°, respectively (Electron Microscopy Data Bank code EMD-22169).

**NMR measurements**. ssNMR measurements on FUS-LC-C and FUS-LC-N fibrils were performed at 17.5 T (745.6 MHz $^1$H NMR frequency), using a Varian InfinityPlus spectrometer console and a 3.2 mm MAS NMR probe from Black Fox, Inc. (Tallahassee, Florida), with MAS frequencies of 12.00 kHz (FUS-LC-C) or 16.50 kHz (FUS-LC-N). The spectrometer was controlled by Spinsight 4.3.2 software. 2D $^{13}$C-$^{13}$C spectra were acquired with 25 ms $^{13}$C-$^{13}$C dipolar-assisted rotational resonance (DARR) mixing periods[71] and 85 kHz two-pulse phase-modulated (TPPM) $^1$H decoupling[72]. Maximum $t_1$ periods were 6.40 ms or 3.20 ms, with 1.5 s recycle delays and total data acquisition times of 10 or 18 h, for FUS-LC-N or FUS-LC-C fibrils, respectively. 2D NCACX spectra were acquired with 4.0 ms $^{15}$N-$^{13}$C$_\alpha$ cross-polarization periods, 25 ms $^{13}$C-$^{13}$C DARR mixing periods, 85 kHz TPPM decoupling, maximum $t_1$ periods of 7.5 ms or 7.9 ms, and total data acquisition times of 23 or 60 h.

The $^{13}$C-detected 2D $^1$H-$^{13}$C INEPT spectrum of FUS-LC-C fibrils was acquired at 17.5 T with 12.00 kHz MAS, using INEPT transfer periods $\tau_1 = 1.25$ ms and $\tau_2 = 0.7$ ms, a maximum $t_1$ period of 10.0 ms, a recycle delay of 2.0 s, and a total data acquisition time of 4.2 h. $^1$H decoupling used a train of 180° pulses with a 25-step phase pattern[73], with a 25 μs 180° pulse length. 1D INEPT spectra with various combinations of $\tau_1$ and $\tau_2$ were acquired in 0.15 h each.

ssNMR measurements on full-length FUS-LC fibrils were performed at 14.1 T (599.1 MHz $^1$H NMR frequency), using a Varian InfinityPlus spectrometer console and a 3.2 mm MAS NMR from Varian. The MAS frequency was 12.00 kHz, TPPM decoupling fields were 85 kHz, $^{13}$C-$^{13}$C spin diffusion mixing periods were 50 ms, and recycle delays were 1.5 s. The 2D $^{13}$C-$^{13}$C spectrum was acquired in 22 h with a maximum $t_1$ period of 6.0 ms. The 2D NCACX spectrum was acquired in 36 h with a maximum $t_1$ period of 14.9 ms, with a 4.0 ms $^{15}$N-$^{13}$C$_\alpha$ cross-polarization period.

Sample temperatures were 24 ± 4 °C in all NMR measurements. 2D spectra were processed with nmrPipe software[74], using 0.4–0.5 p.p.m. Gaussian apodization in both dimensions of $^{13}$C-$^{13}$C spectra and NCACX spectra of FUS-LC-N and full-length FUS-LC fibrils, 0.2 p.p.m. in the $^{13}$C dimensions, 0.5 p.p.m. in the $^{15}$N dimension of $^{13}$C-$^{13}$C and NCACX spectra of FUS-LC-C fibrils, and 0.01 p.p.m. in the $^1$H dimension and 0.2 p.p.m. in the $^{13}$C dimension of $^1$H-$^{13}$C INEPT spectra. 2D spectra were plotted with Sparky software (https://www.cgl.ucsf.edu/home/sparky).

**Development of molecular model**. As described in the main text, the bulky side chains in the density map (Fig. 3) indicated that the Gly-rich C-terminal segment of FUS-LC-C is not included in the fibril core. After examining various possibilities, we found that residues 112–150 fit well into the density map, whereas attempts to fit other segments into the density map resulted in empty side-chain densities or side chains of the model that projected outside the density (Supplementary Fig. 5). A single chain consisting of residues 112–150 was manually fit into the density for one molecule using Coot software[37]. Eight copies of this chain were then fit into the density map for four repeats of the $2_1$-symmetric dimer, using the Fit in Map function of UCSF Chimera[75].

Further refinement of the molecular model was then performed with Xplor-NIH software[38], using the probDistPot potential energy function to restrain all non-hydrogen atoms of the eight chains within the cryo-EM density map. Non-crystallographic symmetry (NCS potential of Xplor-NIH) was included to make the conformations of all chains nearly identical. Translational symmetry (DistSymmPot) restraints were applied to enforce translational symmetry within each of the two cross-β subunits. A PosDiffPot potential was used to restrain the backbone atoms of each chain to within 2.5 Å of their initial positions, in order to prevent each chain from hopping to the density of a neighboring chain during simulated annealing from high temperatures. Torsion angle (TorsionDB), hydrogen bond (HBDB), and standard potentials to define bond lengths and angles, improper dihedral angles, and atomic radii were used (BOND, ANGL, IMPR, and RepelPot potentials). Scale factors for all potential energy terms are summarized in Table 2.

In the first set of Xplor-NIH calculations, ten independent structures were calculated with a probDistPot scaling factor of 10.0, using annealing from 3000 K to 25 K in 12.5 K temperature decrements, with 1500 steps of torsion angle dynamics at each temperature. Annealing was followed by energy minimization in torsion angle and Cartesian space. The two lowest energy structures without violations were used as starting structures for the second set of calculations, which used the same conditions, but with a probDistPot scaling factor of 5.0. Fourteen

structures without violations (except insignificant violations of the NCS potential) were deposited in the Protein Data Bank (PDB code 6XFM) and are shown in Supplementary Fig. 7a.

Another set of calculations (with the same two starting structures) was performed without the TorsionDB potential, to examine how the cryo-EM density map alone determines side-chain conformations. These calculations produced eight structures without violations, shown in Supplementary Fig. 7b.

**MD simulations**. MD simulations were performed with NAMD software[47] and CHARMM22 force fields[76]. Five repeats of a $2_1$-symmetric FUS-LC-C dimer were generated from the lowest energy structure from the first set of Xplor-NIH calculations. The resulting 10-mer was then solvated and neutralized in 100 mM NaCl in a 84.3 Å × 65.0 Å × 46.3 Å box in VMD[77]. Constant-pressure simulations with Langevin dynamics were run at 303 K with periodic boundary conditions. One MD trajectory was calculated for 400 ns, with C$_\alpha$ atoms being constrained to their initial positions by harmonic potentials. A constraint scaling factor of 0.3 was chosen to allow moderate fluctuations of backbone coordinates in this trajectory, corresponding to a backbone RMSD of about 0.8 Å. Another MD trajectory was calculated for 500 ns, without constraints of any type.

Analyses of the MD trajectories were performed with VMD software[77]. Using the labels $A_1$–$A_5$ for molecules in one cross-β subunit and $B_1$–$B_5$ for molecules in the other subunit, data in Supplementary Fig. 8b were obtained by counting the number of water molecules within 7.0 Å of side-chain OH of Y122 or Y143 or within 7.0 Å of side-chain C$_\delta$ of Q132 or Q145 and averaging the resulting numbers for $A_3$ and $B_3$ (in the unconstrained trajectory). For water near the Q133, we counted the number of water molecules within 5.0 Å of side-chain C$_\delta$ of Q133 to exclude water molecules outside the fibril core. For Supplementary Fig. 8c, d, water molecules within 7.0 Å of side-chain OH or C$_\delta$ of Y122, Y143, Q132, and Q145 were selected at arbitrary time points t with the first 10 ns of the unconstrained trajectory.

For Supplementary Fig. 9, hydrogen bonds were defined by donor–acceptor distances <3.0 Å and deviations from linearity less than 20°. Occupancies of hydrogen bonds were defined as the fraction of frames where the hydrogen bonds were found. Data in Supplementary Fig. 9a were averaged over results for $A_3$ and $B_3$. For Supplementary Fig. 9b, occupancies of hydrogen bonds between residues 112–129 and residues 131–150 were also averaged over values for $A_3$ and $B_3$. For Supplementary Fig. 9c, hydrogen bonds between $A_3$ and $B_2$, $B_3$, or $B_4$ and between $B_3$ and $A_2$, $A_3$, or $A_4$ were examined, and occupancies for the two cases were averaged. For Supplementary Fig. 9d, e, hydrogen bonds between $A_3$ and $A_2$ or $A_4$ and between $B_3$ and $B_2$ or $B_4$ were examined, and occupancies for the two cases were averaged.

For Supplementary Figs. 10 and 11, time-dependent torsion angles $\chi_2$ of Tyr residues or pseudo-torsion angles $\xi$ of Gln residues in $A_3$ and $B_3$ were obtained from the unconstrained and constrained trajectories, respectively. The definition of $\xi$ is given in the caption to Supplementary Fig. 10.

**Reporting summary**. Further information on experimental design is available in the Nature Research Reporting Summary linked to this paper.

## Data availability
The cryo-EM density map for FUS-LC-C fibrils is available from the Electron Microscopy Data Bank, code EMD-22169. Atomic coordinates for a bundle of 14 molecular structures that fit the density map are available from the Protein Data Bank, code 6XFM. The density map and coordinates are available at https://doi.org/10.2210/pdb6XFM/pdb. The 2D ssNMR spectra are available at https://doi.org/10.17632/ts9p355m3d.1. All other data are available from the authors upon request.

## Code availability
Modified RELION 3.0 software and associated scripts for cryo-EM image processing are available from the authors upon request.

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

## Acknowledgements

This work was supported by the Intramural Research Program of the National Institute of Diabetes and Digestive and Kidney Diseases, National Institutes of Health. Cryo-EM data were obtained at the NIH Multi-Institute Cryo-EM Facility (MICEF). Calculations of density maps and molecular models were performed on the NIH High Performance Computing Biowulf cluster.

## Author contributions

M.L. and M.K. prepared protein and fibril samples. M.L., U.G., and K.R.T. acquired and analyzed cryo-EM data. M.L., M.K., and R.T. acquired and analyzed NMR data and performed other measurements. M.L. and R.T. wrote the manuscript.

## Competing interests

The authors have no competing interests.
