## [Peer Review File · Nature Communications]

REVIEWER COMMENTS

Reviewer #1 (Remarks to the Author):

The manuscript from Robert and co-workers reports the structural model of FUS-LC-C and discovered a fibril core in this segment other than the known one in the N-terminal. By using multiple technologies like ssNMR, cryo-EM and computational stimulation, the authors provide detailed structural information of the fibril formed by FUS-LC-C. The fibril displays clear twist structure composed by two cross-beta subunits. Additionally, they made comparison between the structures of FUS-LC-N and -C in order to explain why the FUS-LC-C core does not exist in full FUS-LC fibrils. Overall, the report presents nice data and comprehensive discussion, which gives us some insights on structural basis of FUS self-assembling. However, some issues listed below should be addressed before publication.

1. As mentioned in the manuscript and known to public, the full C-terminal of FUS-LC is highly disordered. Herein the authors truncated the C-terminal to 111-214 and found fibril aggregation occurred. How the sequence was determined? Will a slightly longer or shorter segment significantly change the aggregation status? A discussion will help to support the hypothesis that segments outside the fibril core prevent the C-terminal core formation in full LC.
2. The authors mentioned a lot of phase separation in the introduction. People have known that the disordered segment makes great contribution to LLPS. However, it is difficult to characterize structures of proteins under phase separation status. Will the findings here provide any insights on phase separation, or the conversion process from LLPS to fibril?
3. Since the LC-C fragment does not naturally exist and the structure is different from the full FUS, I am a little concerned about the significance of the findings here. Is the C-terminal core related to pathogenesis? Or can a universal model for self-assembling proteins be established from the analysis of LC-C sequence or structure? A further discussion will help to improve the significance a lot.
4. In second paragraph of the introduction, it should be "FUS (Fused in Sarcoma)" not "FUUsed".

Reviewer #2 (Remarks to the Author):

Myungwoon Lee et al reported the crystal structure and interaction mode of amyloid fibril for FUS. In the FUS-LC-C fibril core, residues 112-150 adopt U-shaped conformations and form two subunits

with in-register, parallel cross-beta structures, arranged with quasi-21 symmetry.

All-atom molecular dynamics simulations indicate that the FUS-LC-C fibril core is stabilized by a plethora

of hydrogen bonds involving sidechains of Gln, Asn, Ser, and Tyr residues, both along and transverse to

the fibril growth direction, including diverse sidechain-to-backbone, sidechain-to-sidechain, and sidechain-to-water

interactions. These results give a image about the structure and function of fibril. Therefore, this is a good job. However,

there are also some major points to refine.

1. Please calculate and compare the population of disordered promoting residues between FUS-LC-C and FUS-LC-N.

Then the reader will find the propensity of disordered sequence and understand the sequence character of fibril.

2. For the simulation system, it is necessary to evaluate the convergence of simulation system.

3. For the aggregation of amyloid fibril, the native contact and salt bridge also play key roles in the stability. Therefore,

please discuss this point.

Reviewer #3 (Remarks to the Author):

The work "Molecular Structure and Interactions Within Amyloid-Like Fibrils Formed by a Low-Complexity Protein Sequence from FUS" by Lee et al. is a comprehensive and beautifully presented work representing the structural model of FUS in different forms.

Cryo EM, combined with qualitative solid-state NMR spectral analysis by the authors utilised for the conclusions made in the work.

I find the work very relevant to the community, timely, and solid, and fits to be published.

FUC-LC is a rather interesting protein, with a low complexity. I am not sure if I have seen any other fibril before having so many G, S and so on. Very interesting, in terms of structural studies, however the same factor makes it very difficult to analyse the solid-NMR spectra due to overlap, and very likely repeated patterns.

The preparation and analysis of the FUS-LC-C (111-214), FUS-LC-N (1-110), and FUS-LC (1-214) samples, is making the solid-nmr more appealing, due to the disentanglement of several issues intrinsic to the system.

I find it amusing that the in-vitro studies on the soluble LC domain, resulted in a "hydrogel" containing entangled fibrils / precipitated fibril bundles. This reminds the other interesting case of TasA protein, recently studied, in which case the similar procedure resulted in "oligomeric" species in a similar treatment (whereas fibrils or monomers are produced under different conditions).

The Solid-state NMR spectra looks in general adequate and promising for analysis & further analysis, with narrow linewidths, although very overlap due to "LC".

Here are my comments/questions:

- First of all, I am very interested in seeing the biophysical characterisation on the samples, including the interesting preparation I mentioned above.

It is extremely important to differentiate the "fibril" species from any possibly forming "oligomers". This is crucial for the analysis.

- I am wondering if the mentioned mutations on page 3, are occurring in the LC-N or LC-C domains in the protein? This information could be useful for the analysis and discussion.

- The 2D spectra shown in Fig. Figure 1 & Figure SI1 presented nicely. I am not sure how the contour levels in SI1 set. Only the contour-spacing is mentioned. It would be nice to mention the base-level, and how they are adjusted for a fair comparison. Similarly also on Figure 1 @NC spectra.

- I am not sure if I understood the argument on page 5:

"Crosspeaks positions in 2D spectra of FUS-LC-N fibrils are in good agreement with crosspeak positions in 2D spectra of full-length FUS-LC fibrils (Fig. 1E and S1B). Thus, the molecular structure of the FUS-LC-N fibril core is very similar to the core structure in full-length FUS-LC fibrils determined earlier²¹".

Can Authors be more explicit about the similar / different cross-peaks and their positions in the spectra?

how many cross peaks fit? how many does not?

I am not sure if its safe to propose that the overall folds of the proteins are similar by looking at a few same/similar position cross peaks. I can see many crosspeaks at different positions on spectra of different samples as well.

- I think it would be more complete if the authors cite more recent solid-NMR fibrillar studies, such as the one I mentioned above on TasA from 2018 I guess, as well as another more recent ssNMR / functional amyloid work on FapC. This would be highly positive for the completeness of the story, since the studied system resembles more the functional-amyloids, rather than the disease making fibrils.

- Would I be too unfair if I ask about a rough cross-peak count on the spectra ? And a discussion of this along with the polymorphism ? I counted 27S & 18G in the LC-N, and correspondingly on the LC-C. At least if the authors would comment on this, it would be very kind. Maybe a more quantitative aminoacid-type would be selected, if there is such a case.

Overall a very nice work.
kind Regards

Reviewer #4 (Remarks to the Author):

This manuscript describes a cryo-EM structure of an amyloid fibril formed in vitro from a recombinantly-expressed fragment of FUS protein comprising the C-terminal half of the low complexity domain (residues 111-214).

The structure reveals a U-shaped ordered core formed by residues 112-150, which forms left-handed fibrils comprising two protofilaments, and is supported by NMR, AFM and dark-field TEM measurements. Molecular dynamics simulations using the structure support the presence of water molecules within the ordered core; sidechain-to-backbone and sidechain-to-sidechain hydrogen bonding; and dynamic sidechain conformations. NMR analysis of amyloid fibrils formed from a different fragment of FUS protein comprising the N-terminal half of the low complexity domain (residues 2-108) reveal a different structure that is consistent with that formed by a fragment containing the entire low complexity domain and previously published in (Murray et al. 2017 Cell).

The physiological and pathophysiological relevance of the work presented here is not clear. The physiological roles of FUS are carried out by the full-length protein and require additional domains. Moreover, a role for amyloid formation by full-length FUS in LLPS or other physiological processes has not been shown. In neurodegenerative diseases, intracellular aggregates are also comprised of full-length FUS protein (Neumann et al. 2009 Brain) and do not stain with the amyloid-binding dye Thioflavin-S (Bigio et al. 2013 Acta Neuropathologica). This manuscript will, therefore, mostly appeal to biophysicists studying the structural basis of amyloid polymorphism.

The structural and other experimental data on these amyloid fibrils appear to be sound and enable interesting comparisons to be made with published work on other amyloid fibrils formed by FUS, as well as other proteins. Such structures and comparisons are sorely needed by the community to elucidate the structural basis of amyloid formation and polymorphism.

One minor comment- at 2.62 Å resolution, one might expect to see evidence of ordered solvent molecules and alternative sidechain conformations in the cryo-EM 3D reconstruction. Is this the case and do they support the molecular dynamics simulations?

The accompanying files are a revised version of manuscript NCOMMS-20-24782. The original version received four reviews, all of which appear to be generally favorable. The four reviewers requested a variety of revisions and clarifications. We thank the reviewers for their time and their helpful comments. To address their comments, we have made numerous revisions to the text and figures. The reviews appear below in red. Our responses and revisions appear in blue:

Reviewer #1 (Remarks to the Author):

The manuscript from Robert and co-workers reports the structural model of FUS-LC-C and discovered a fibril core in this segment other than the known one in the N-terminal. By using multiple technologies like ssNMR, cryo-EM and computational stimulation, the authors provide detailed structural information of the fibril formed by FUS-LC-C. The fibril displays clear twist structure composed by two cross-beta subunits. Additionally, they made comparison between the structures of FUS-LC-N and -C in order to explain why the FUS-LC-C core does not exist in full FUS-LC fibrils. Overall, the report presents nice data and comprehensive discussion, which gives us some insights on structural basis of FUS self-assembling. However, some issues listed below should be addressed before publication.

1. As mentioned in the manuscript and known to public, the full C-terminal of FUS-LC is highly disordered. Herein the authors truncated the C-terminal to 111-214 and found fibril aggregation occurred. How the sequence was determined? Will a slightly longer or shorter segment significantly change the aggregation status? A discussion will help to support the hypothesis that segments outside the fibril core prevent the C-terminal core formation in full LC.

Response: As explained in the Introduction to our manuscript, we have shown in previous publications that fibrils formed by FUS-LC (the low-complexity sequence consisting of residues 2-214 of the full-length FUS protein) have a structurally ordered core formed by residues 39-95, and that most of the C-terminal half of FUS-LC remains dynamically disordered when FUS-LC forms fibrils. In the current manuscript, we study fibrils formed by residues 111-214, which we call FUS-LC-C. This is nearly the full dynamically disordered segment of FUS-LC fibrils. We also report solid state NMR data for fibrils formed by residues 2-108 of FUS-LC, which we call FUS-LC-N (Figures 1C, 1E, and S1B). We have not studied shorter or longer segments of the C-terminal half of FUS-LC. However, based on other experiments in our lab and papers from other labs (e.g., Luo et al., Nature Struct. Mol. Biol., vol. 25, pp. 341-346, 2018; Ding et al., J. Mol. Biol. vol. 432, pp. 467-483, 2020), we expect that many segments that are rich in Ser, Gln, and Tyr residues will also form amyloid-like fibrils.

As suggested by the reviewer, it will be interesting in future work to examine a series of segments such as residues 2-X, with $X < 214$. As X decreases (i.e., as the disordered C-terminal sequence becomes shorter), the fibril structure may switch from the N-terminal to the C-terminal core. Such experiments will require lengthy solid state NMR or cryoEM measurements and data analyses, and are therefore beyond the scope of the current manuscript.

To address the reviewer's comment, we have added the following sentences on page 14:

"In future experiments on other segments of FUS-LC, it may be possible to demonstrate the

effects of disordered segments on the identities and structures of core-forming segments, using cryo-EM and ssNMR measurements. For example, in fibrils formed by residues 2-X of FUS-LC, the core structure may switch from that of FUS-LC-N to that of FUS-LC-C as X decreases from 214 to 150."

2. The authors mentioned a lot of phase separation in the introduction. People have known that the disordered segment makes great contribution to LLPS. However, it is difficult to characterize structures of proteins under phase separation status. Will the findings here provide any insights on phase separation, or the conversion process from LLPS to fibril?

Response: This is a good question, but we do not have a good answer. As the reviewer says, structures of low-complexity proteins in their liquid-like phase-separated states are difficult to characterize. The FUS-LC-C fibril structure reported in our manuscript does not necessarily have relevance to LLPS. It is possible that FUS-LC molecules adopt similar structures in short-lived, transient, oligomeric assemblies within phase-separated "droplets". However, we do not want to discuss this possibility in our manuscript without experimental evidence.

3. Since the LC-C fragment does not naturally exist and the structure is different from the full FUS, I am a little concerned about the significance of the findings here. Is the C-terminal core related to pathogenesis? Or can a universal model for self-assembling proteins be established from the analysis of LC-C sequence or structure? A further discussion will help to improve the significance a lot.

Response: The results in our manuscript are significant primarily from the biophysical perspective. We do not claim that the FUS-LC-C core structure reported in our manuscript exists within pathogenic FUS fibrils (although this is a possibility). However, this structure and our molecular dynamics simulations based on this structure provide new information about the variety of fibril structures that low-complexity sequences can adopt and about the nature of interactions that stabilize these structures. In addition, the final paragraph of the Discussion section already discusses the generality of the FUS-LC-C and FUS-LC-N structural motifs.

To address this comment, the following sentence on page 4:

"Although we do not claim that the FUS-LC-C fibril structure discussed below has direct relevance to pathogenesis or biological function, this work provides new information about the diversity of structural motifs that can exist within low-complexity protein fibrils, their relation to motifs in fibrils formed by other types of sequences, and the diversity of intermolecular and inter-residue interactions that can stabilize amyloid fibril structures."

4. In second paragraph of the introduction, it should be "FUS (Fused in Sarcoma)" not "FUsed".

Response: We have made this change in the text.

Reviewer #2 (Remarks to the Author):

Myungwoon Lee et al reported the crystal structure and interaction mode of amyloid fibril for FUS. In the FUS-LC-C fibril core, residues 112-150 adopt U-shaped conformations and form two subunits with in-register, parallel cross-beta structures, arranged with quasi-21 symmetry. All-atom molecular dynamics simulations indicate that the FUS-LC-C fibril core is stabilized by a plethora of hydrogen bonds involving sidechains of Gln, Asn, Ser, and Tyr residues, both along and transverse to the fibril growth direction, including diverse sidechain-to-backbone, sidechain-to-sidechain, and sidechain-to-water interactions. These results give a image about the structure and function of fibril. Therefore, this is a good job. However, there are also some major points to refine.

1. Please calculate and compare the population of disordered promoting residues between FUS-LC-C and FUS-LC-N. Then the reader will find the propensity of disordered sequence and understand the sequence character of fibril.

Response: To address this comment we have added the following sentences on page 15:

"It should also be noted that FUS-LC-N contains 18 Gly and 7 Pro residues, while FUS-LC-C contains 36 Gly and 4 Pro residues. The larger population of Gly residues in FUS-LC-C may contribute to the identity of the core-forming segment in FUS-LC fibrils. The fact that Gly residues are concentrated after residue 165 contributes to the selection of residues 112-150 for the core of FUS-LC-C fibrils."

2. For the simulation system, it is necessary to evaluate the convergence of simulation system.

Response: In a revised version of Figure S8, we have added plots of the RMSD for backbone atoms as a function of the simulation time. These plots show that the fibril structure converges within about 10 ns in our molecular dynamics simulations that include backbone constraints, and within about 50 ns in simulations that do not include backbone constraints.

3. For the aggregation of amyloid fibril, the native contact and salt bridge also play key roles in the stability. Therefore, please discuss this point.

Response: To address the reviewer's comment, we have added the following sentences on page 16:

"Hydrophobic interactions and salt bridges can also be important stabilizing interactions for pathogenic amyloid fibrils, as indicated by structural studies of amyloid- β and α -synuclein fibrils, as well as for functional amyloids. The absence of charged amino acid sidechains and the absence of purely hydrophobic residues other than Pro within the FUS-LC-C fibril core implies that hydrophobic interactions and salt bridges do not account for the formation of FUS-LC-C fibrils."

(Note that the phrase "native contact" usually means inter-residue contacts that exist in the folded state of a protein. There is no folded state for monomeric FUS-LC, so this is not a relevant

concept. I assume that the reviewer means "hydrophobic contact".)

Reviewer #3 (Remarks to the Author):

The work "Molecular Structure and Interactions Within Amyloid-Like Fibrils Formed by a Low-Complexity Protein Sequence from FUS" by Lee et al. is a comprehensive and beautifully presented work representing the structural model of FUS in different forms.

Cryo EM, combined with qualitative solid-state NMR spectral analysis by the authors utilised for the conclusions made in the work.

I find the work very relevant to the community, timely, and solid, and fits to be published.

FUC-LC is a rather interesting protein, with a low complexity. I am not sure if I have seen any other fibril before having so many G, S and so on. Very interesting, in terms of structural studies, however the same factor makes it very difficult to analyse the solid-NMR spectra due to overlap, and very likely repeated patterns. The preparation and analysis of the FUS-LC-C (111-214), FUS-LC-N (1-110), and FUS-LC (1-214) samples, is making the solid-nmr more appealing, due to the disentanglement of several issues intrinsic to the system.

I find it amusing that the in-vitro studies on the soluble LC domain, resulted in a "hydrogel" containing entangled fibrils / precipitated fibril bundles. This reminds the other interesting case of TasA protein, recently studies, in which case the similar procedure resulted in "oligomeric" species in a similar treatment (whereas fibrils or monomers are produced under different conditions).

The Solid-state NMR spectra looks in general adequate and promising for analysis & further analysis, with narrow linewidths, although very overlap due to "LC".

Here are my comments/questions:

- First of all, I am very interested on seeing the biophysical characterisation on the samples, including the interesting preparation I mentioned above.

It is extremely important to differentiate the "fibril" species from any possibly forming "oligomers". This is crucial for the analysis.

Response: Negative-stain TEM images and cryoEM images (see Figs. 1 and 2) show that the FUS-LC-C fibril preparation protocol described on page 3 of the Supplementary Information produces samples that are primarily fibrillar. A small fraction of amorphous-looking aggregates are also seen, but these are easily distinguished from fibrils in the cryoEM images and therefore do not contribute to the cryoEM density map or to the molecular structural model that we derived from the density map.

On page 13, we also report the equilibrium solubility of FUS-LC-C under our experimental conditions. The value of 4.5 μM indicates that soluble oligomeric species (i.e., oligomers that are too small to be pelleted in our solubility measurements) do not constitute a large fraction of our samples.

To address the reviewer's comment, we have added the following sentence in the Materials and Methods section (page 3 of the Supplementary Information):

"Approximately 20% of our cryo-EM images of FUS-LC-C fibrils contain minor fractions of non-fibrillar aggregates. Since non-fibrillar aggregates are readily distinguished from fibrils in the manual particle picking process, they do not contribute to the density map or structural model for FUS-LC-C fibrils."

- I am wondering if the mentioned mutations on page 3, are occurring in the LC-N or LC-C domains in the protein ? This information could be useful for the analysis and discussion.

Response: Most disease-associated mutations that have been identified in the low-complexity domain of FUS do indeed occur in the FUS-LC-C segment. We have added the following sentence on page 4, along with a reference to a relevant review article:

"The majority of disease-associated mutations in the low-complexity domain of FUS do occur after residue 111."

- The 2D spectra shown in Fig. Figure 1 & Figure S11 presented nicely. I am not sure how the contour levels in S11 set. Only the contour-spacing is mentioned. It would be nice to mention the base-level, and how they are adjusted for a fair comparison. Similarly also on Figure 1 @NC spectra.

Response: 2D spectra of different samples in Figures 1 and S1 are plotted with the same contour level increments and approximately the same number of contours below the maximum crosspeak signals. This ensures a fair comparison of similarities and differences, even when the signal-to-noise ratios in the 2D spectra may vary. We have changed the captions of Figures 1 and S1 to read:

"Contour levels in all 2D spectra increase by successive factors of 1.3 (or 1.5 in Fig. S1) and were set to show approximately the same number of levels below the maximum crosspeak signals in all spectra."

- I am not sure if I understood the argument on page 5:

"Crosspeaks positions in 2D spectra of FUS-LC-N fibrils are in good agreement with crosspeak positions in 2D spectra of full-length FUS-LC fibrils (Fig. 1E and S1B). Thus, the molecular structure of the FUS-LC-N fibril core is very similar to the core structure in full-length FUS-LC fibrils determined earlier²¹."

Can Authors be more explicit about the similar / different cross-peaks and their positions in the spectra?

how many cross peaks fit ? how many doesnot ?

I am not sure if its safe to propose that the overall folds of the proteins are similar by looking at a few same/similar position cross peaks. I can see many crosspeaks at different positions on spectra of different samples as well.

Response: It seems clear that 2D spectra of FUS-LC-N and full-length FUS-LC fibrils are very similar to one another in Figures 1E and S1B. Vertical dashed lines in these figure panels show that a large number of crosspeaks (not just a few) do indeed have identical positions (i.e., differences less than the linewidths). Given the well-known sensitivity of ^{13}C and ^{15}N chemical shifts to molecular structure, our conclusion that FUS-LC-N and FUS-LC fibril core structures are very similar seems unassailable. To address this comment, we have added the following sentences on page 5:

"Specifically, 35 resolved or partially resolved crosspeaks in the 2D ^{15}N - ^{13}C spectrum of FUS-LC-N fibrils have chemical shift values that agree to within the ssNMR linewidths with those of crosspeaks in the corresponding spectrum of FUS-LC fibrils (Fig. 1E). More than 50 resolved or partially resolved crosspeaks in the 2D ^{13}C - ^{13}C spectrum of FUS-LC-N fibrils have chemical shift values that agree to within the ssNMR linewidths with those of crosspeaks in the corresponding spectrum of FUS-LC fibrils (Fig. S1B)."

- I think it would be more complete if the authors cite more recent solid-NMR fibrillar studies, such as the one I mentioned above on TasA from 2018 I guess, as well as another more recent ssNMR / functional amyloid work on FapC. This would be highly positive for the completeness of the story, since the studied system resembles more the functional-amyloids, rather than the disease making fibrils.

Response: The TasA and FapC amino acid sequences do not resemble the amino acid sequence of FUS-LC. To my knowledge, structures of TasA and FapC fibrils have not been reported, from either solid state NMR or cryo-EM measurements. Therefore, it is unclear whether TasA or FapC fibrils resemble the FUS-LC and FUS-LC-C fibrils discussed in our manuscript.

As requested by the reviewer, we have added references to recent papers by Loquet and coworkers (FASEB J., vol. 33, pp. 12146-12163, year 2019) and Akbey and coworkers (J. Mol. Biol., vol. 432, pp. 2232-2252, year 2020), about TasA and FapC, respectively. These references appear on page 16, in the sentence "Hydrophobic interactions and salt bridges can also be important stabilizing interactions for pathogenic amyloid fibrils, as indicated by structural studies of amyloid- β and α -synuclein fibrils, as well as for functional amyloids."

- Would I be too unfair if I ask about a rough cross-peak count on the spectra ? And a discussion of this along with the polymorphism ? I counted 27S & 18G in the LC-N, and correspondingly on the LC-C. At least if the authors would comment on this, it would be very kind. Maybe a more quantitative aminoacid-type would be selected, if there is such a case.

Response: To address this comment, we have added the following sentences on page 9:

"Thirteen Ser residues are contained within the core-forming segment of FUS-LC-C, in rough agreement with the number of resolved or partially resolved Ser C_β crosspeaks in Fig. 1D. However, only six Gly residues are contained within the core-forming segment, whereas the 2D ^{15}N - ^{13}C ssNMR spectrum shows signals from at least 14 inequivalent Gly residues. This observation is consistent with the conclusion from INEPT spectra that only a subset of the Gly

residues are dynamically disordered."

Reviewer #4 (Remarks to the Author):

This manuscript describes a cryo-EM structure of an amyloid fibril formed in vitro from a recombinantly-expressed fragment of FUS protein comprising the C-terminal half of the low complexity domain (residues 111-214).

The structure reveals a U-shaped ordered core formed by residues 112-150, which forms left-handed fibrils comprising two protofilaments, and is supported by NMR, AFM and dark-field TEM measurements. Molecular dynamics simulations using the structure support the presence of water molecules within the ordered core; sidechain-to-backbone and sidechain-to-sidechain hydrogen bonding; and dynamic sidechain conformations. NMR analysis of amyloid fibrils formed from a different fragment of FUS protein comprising the N-terminal half of the low complexity domain (residues 2-108) reveal a different structure that is consistent with that formed by a fragment containing the entire low complexity domain and previously published in (Murray et al. 2017 Cell).

The physiological and pathophysiological relevance of the work presented here is not clear. The physiological roles of FUS are carried out by the full-length protein and require additional domains. Moreover, a role for amyloid formation by full-length FUS in LLPS or other physiological processes has not been shown. In neurodegenerative diseases, intracellular aggregates are also comprised of full-length FUS protein (Neumann et al. 2009 Brain) and do not stain with the amyloid-binding dye Thioflavin-S (Bigio et al. 2013 Acta Neuropathologica). This manuscript will, therefore, mostly appeal to biophysicists studying the structural basis of amyloid polymorphism.

Response: As also explained in our response to reviewer #1, we agree that the results in our manuscript are significant primarily from the biophysical perspective. In our manuscript, we do not claim otherwise.

The structural and other experimental data on these amyloid fibrils appear to be sound and enable interesting comparisons to be made with published work on other amyloid fibrils formed by FUS, as well as other proteins. Such structures and comparisons are sorely needed by the community to elucidate the structural basis of amyloid formation and polymorphism.

One minor comment- at 2.62 Å resolution, one might expect to see evidence of ordered solvent molecules and alternative sidechain conformations in the cryo-EM 3D reconstruction. Is this the case and do they support the molecular dynamics simulations?

Response: We do see density attributable to ordered water molecules in our cryoEM-based density map. We now indicate this density with asterisks in Figure 3C. We have added the following sentence on page 11:

"Density attributable to partially ordered water molecules is observed near sidechains of Q140 and S135 in the cryo-EM density map (Fig. 3C), consistent with the MD simulations."

REVIEWERS' COMMENTS

Reviewer #1 (Remarks to the Author):

The authors have adequately addressed all of my concerns.

Reviewer #2 (Remarks to the Author):

No more comments.

Reviewer #3 (Remarks to the Author):

Authors answered my questions.
I am satisfied.

suggest the publication of the article.